# Aesthetic appraisals of literary style and emotional intensity in narrative engagement are neurally dissociable

Franziska Hartung [1,5,6]✉, Yuchao Wang [1,2,6], Marloes Mak[3], Roel Willems[3,4] & Anjan Chatterjee[1]

Humans are deeply affected by stories, yet it is unclear how. In this study, we explored two aspects of aesthetic experiences during narrative engagement - literariness and narrative fluctuations in appraised emotional intensity. Independent ratings of literariness and emotional intensity of two literary stories were used to predict blood-oxygen-level-dependent signal changes in 52 listeners from an existing fMRI dataset. Literariness was associated with increased activation in brain areas linked to semantic integration (left angular gyrus, supramarginal gyrus, and precuneus), and decreased activation in bilateral middle temporal cortices, associated with semantic representations and word memory. Emotional intensity correlated with decreased activation in a bilateral frontoparietal network that is often associated with controlled attention. Our results confirm a neural dissociation in processing literary form and emotional content in stories and generate new questions about the function of and interaction between attention, social cognition, and semantic systems during literary engagement and aesthetic experiences.

[1] Penn Center for Neuroaesthetics, University of Pennsylvania, Philadelphia, PA, USA. [2] Haverford College, Haverford, PA, USA. [3] Center for Language Studies, Radboud University, Nijmegen, Netherlands. [4] Donders Institute for Brain, Cognition, and Behaviour, Radboud University, Nijmegen, Netherlands. [5] Present address: School of Psychology, Newcastle University, 4th Floor Dame Margaret Barbour Building Wallace Street, Newcastle upon Tyne NE2 4DR, UK. [6] These authors contributed equally: Franziska Hartung, Yuchao Wang. ✉email: franziska.hartung@newcastle.ac.uk

Aesthetic emotions are a controversial and elusive concept[1–3] (see review in refs. [4–6]). While some argue that aesthetic emotions are a distinct class of emotions that are behaviourally and biologically distinguishable from other emotions[1,6], empirical evidence to support this claim that aesthetic emotions are distinguishable from other affective states is lacking (see review in ref. [4]). The process of aesthetics evaluation has subjective and cultural components both of which are affected by stimulus properties, perceiver preferences, self-relevance, and a dynamic interaction between stimulus and perceiver as well as the interaction between different systems within the perceiver and layers of a given stimulus[6]. Aesthetic experiences are thought to be an "emotionally and hedonically engaging, conscious experience of an aesthetic quality of a stimulus" and "arise from: (i) an aesthetic evaluation (i.e., the perception and higher-order processing of stimulus properties that are diagnostic of an aesthetic quality); (ii) a felt emotional component; and (iii) felt aesthetic (dis)pleasure"[6].

Aesthetic emotions cannot easily be assessed by valence and arousal. For instance, the feeling of being moved is neither positive or negative, and has high emotional intensity while being low in arousal[7,8]. It is unclear whether aesthetic experiences are supported by domain general brain systems such as the reward system or whether different aspects or types of aesthetic emotions are supported by domain-specific brain systems. In this study, we explore aesthetic experiences when people engage with literary narratives. We investigate whether two different aspects of aesthetic engagement—appreciation of stylistic form of literary language and emotional intensity of fluctuations in the plot (e.g., suspense) are neurally dissociable or share common activations (e.g., in reward systems). This exploratory study further aims to generate hypotheses for neuroimaging and behavioural research on literary aesthetics and other narrative art traditions (e.g., films, theatre).

Narratives fluctuate over their durations in the literary language used and emotional intensity (suspense) conveyed. They are thought to be "the human brain's way of consolidating and conveying the temporally evolving world we live in"[9]. Because of the dynamic nature of aesthetic engagement, narratives allow us to explore neural correlates of different aspects of aesthetic engagement that unfold over time. Previous research has broadly categorized affective responses to narratives into narrative emotions and emotions related to aesthetic experiences of the literary form (e.g., style of the writing, building of the plot structure). Narrative emotions are elicited when recipients are transported or immersed into the narrative world, e.g., the joy when someone projects themselves in the protagonist's situation (emotions of empathy) or the suspense they feel when a carefree protagonist is happily walking in the woods, while the readers (or listeners or viewers) are keenly aware of looming danger (emotions of sympathy)[10,11]. In contrast, aesthetic responses to the writing or narrating itself are based on meta-features such as the style and place of a literary tradition, the narrative and figurative devices used, and the plot construction[10,12].

While not everybody can write literature, most people have intuitions about what makes a text literary[13–16]. Mukařovský[17] (see also ref. [18]) proposed the idea of foregrounding, which entails the "systematic employment of a range of stylistic devices" such that literary language "becomes more conscious to the reader compared to language of standard spoken discourse or informative text." For instance, reading "shall I compare thee to a summer's day?" from Shakespeare's sonnets makes one aware of its language both because of the analogy offered and its archaic style.

Mukařovský[17] proposes three levels of foregrounding: (i) the phonetic level (e.g., alliteration, rhyme), (ii) the grammatical level (e.g., inversion and ellipsis), and (iii) the semantic level (e.g., metaphors, irony). Crucially, foregrounding is always in relation to a given context. For instance, the context for a novel would be the text itself, what the recipient knows about the writer and their other works, including fan fiction, and the placement in literature (e.g., time, genre, style). Segments of a text are foregrounded, while others are backgrounded. The text and its context itself define segments that stand out and segments that are perceived as the background. In a story using a lot of flowery language, colloquial expressions might stand out, whereas in a text written in very clear report style language, a metaphor or an emotional word can stand out as foregrounded. While everyday language also contains foregrounding and stylistic features used in literature, the quality that differentiates a piece of literature from our daily discourse is the systematic use of such features in conjunction with skilful narration.

The hypothesized effects of foregrounding have been assessed empirically[7,14,15,19–21]. In cognitive terms, foregrounding produces a prediction violation that results from the context of the text itself and local use of stylistic markers[22]. Foregrounding—by definition—deviates from the norm and hence expectations based on the backgrounded narrative language within a given context. The brain predicts incoming linguistic information during comprehension at all linguistic levels including form (e.g., phonetic features, word form, syntax) and semantic contents (e.g., words, sentence contents) (see review in ref. [23]). Prediction violations occur when linguistic information is not congruent with the reader's expectation and typically produce electrophysiological potentials or increased blood flow in areas linked to language processing showing that cognitive demand is increased (see review in ref. [24] on prediction violation in emotion language processing).

It is important to point out that while prediction violation in psycholinguistics (often related to word frequency in linguistic corpora) and prediction violation in foregrounding in literature may produce similar cognitive effects, the concept of foregrounding is orthogonal to lexical frequency. Foregrounding is defined by probability within a given limited context. In fact, particularly frequent lexical items are often foregrounded in a story. A common example of foregrounding of frequent lexical items is when colloquial language in literary writing is used to contrast a more formal narrative style. Foregrounding is also not restricted to a linguistic level or language units (e.g., lexical items, words, or sentences). Rather foregrounding is a subjective experience of salience in parsing intentionally constructed language for aesthetic purposes. Since there are no agreed upon objective measures for foregrounding, a pragmatic approach of assessing foregrounding with subjective ratings of a group of naive participants and then testing for the reliability across raters is used to study foregrounding empirically.

Foregrounding in language can add cognitive load and increase aesthetic appreciation[14,15,21,25–27]. Moreover, experience or "training" in reading literature seems to be linked to different reading strategies when encountering foregrounded language. Van den Hoven et al.[15] showed that frequent readers of literature tend to accelerate their forward eye movements during reading when encountering literary language. Infrequent readers in contrast are more likely to slow down and regress more often towards preceding language (see also ref. [28]). The slowing down of reading upon encountering foregrounded language further predicted increased appreciation of the literary works, confirming Mukařovský's theory that the disruption of rapid default processing and increased awareness of stylistic features contributes to aesthetic appreciation[15]. Neuroimaging evidence suggested that foregrounding devices such as alliteration modulate attention and semantic integration[29]. Evidence from neuroimaging research

also showed that certain types of literary language (metaphors, idioms, and irony/sarcasm) increase activation in the parietal attention network[30], presumably by increasing processing demands.

In contrast to foregrounding in literary style, backgrounding supports immersive processes like being absorbed in the text and feeling transported into the narrative[25] (but see refs. [31,32]). Immersion during reading literature is connected to engaging socially and emotionally with fictional characters, experiencing the problems and situations communicated in the story and appreciating the social and cultural implications of a story. The extent of immersion and its contribution to aesthetic evaluation is likely affected by individual differences in what type of stories and problems readers find interesting and relevant for themselves.

Previous neuroimaging studies on affective experiences during reading of literature tend to focus on responses such as suspense[33] and story immersion[34,35]. Social and emotional engagement with literary characters has been associated with activations in brain areas linked to affective empathy[34,36–38] as well as social cognition and predictive inference (using real world knowledge and previous context to have an expectation of story development[39]. Empathy with and social cognition about characters in narratives and predictions made may be crucial to producing suspense[39,40]. The neural systems underlying social cognition and empathy are neurally dissociable from the evaluation and reward system[41], but their role in aesthetic appreciation is unclear.

One functional MRI study on the parametric effects of negative valence on neural activations during narrative comprehension showed that increased negative valence correlates with activation in the striatum and amygdala, which are strongly associated with reward and fear processing, as well as parts of the brain network linked to social cognition[42]. Liking of negatively valenced stories was further linked to increased activation in the medio-prefrontal cortex during story engagement[43].

Despite a rich tradition in literary theory and neural evidence regarding emotion word processing and the effect of emotion on lexical processing (see review in refs. [24,44]), it is unclear how aesthetic experiences and aesthetic emotions in narrative processing are neurally supported. Empirical evidence shows that words that are perceived as emotional as opposed to neutral correlate with increased activation in brain areas associated with semantic and lexical processing areas—which might reflect their saliency—and neural systems processing affective information (see review in ref. [24]).

Narrative emotions are complex and cannot be reduced to lexical values of individual words[37,45,46] (however, see ref. [35] for a successful implementation of this approach with popular fiction excerpts). Intensity in stories may arise without any overt emotion depicting words and vice versa emotion words can be neutral and without intensity in narrative contexts[37]. Moreover, arousal and valence for individual words can drastically change between narratives depending on the meaning assigned within the context of the story (narrative world). For instance, an enchanted object presenting luring dangers for the protagonists in a fantasy story can be associated with strong emotions, while the word itself referring to the object in standard vocabulary has low arousal and neutral valence. It is not clear how valence and arousal ratings associated with individual word meanings relate to narrative processing where the contextual embedding alone can produce emotional responses independent of word meaning and can likewise neutralize emotion semantics of words depending on the context.

Similarly, research on empathy with fictional characters (see review in ref. [38]) and Theory of Mind (ToM) or mentalizing[47,48] does not provide a framework that explains or predicts aesthetic

processes outside of experienced reward for simulated (social) experiences and learning. Exploring the complexity of the social cognition relation and communicative intention between readers, narrators, authors, and characters is beyond the scope of this study (see refs. [45,46]). Carefully designed behavioural studies are needed to disentangle these social relations systematically.

In this exploratory approach we focus on global modulations of plot related changes in emotional intensity. This operationalization of intensity is conceptually linked to suspense and arousal and is likely affected by empathy with the characters. Suspense and emotional content in narratives has been shown to modulate attention. In a study on visual narratives, narrative suspense has been shown to suppress attentional focus to peripheral sensory input and deactivation of the default mode network[49]. The default mode network also has been linked to accumulative plot formation[50] (see also ref. [51]). In a fMRI study on narrative emotions during processing of a highly culturalized folk fairytale, Wallentin et al.[37] related independent appraisals of emotional intensity to heart rate variability and increased activation in areas linked to conditioned emotional responses such as the amygdala and thalamus in another group of subjects. Suspense has further been linked to increased activation in brain areas associated with predictive inference and social cognition[39].

The aim of this study was to disentangle the neural bases of aesthetic experience linked to appreciation of the writing of a story from narrative emotions such as suspense and emotional intensity induced by the events in the story. We hypothesized that appreciating literary style and experiencing narrative fluctuations in emotional intensity are supported by different neural systems. Specifically, we hypothesized that literariness, in line with the foregrounding hypothesis, engages neural systems associated with language and attention. We had less specific predictions for brain areas associated with the processing of changes in emotional intensity during story engagement which, in the context of this study, are appraisals of global emotion intensity and arousal (conscious reflection of the emotional experience), rather than felt specific emotions that can be assigned into different emotion categories (physiological emotional states; see discussion in ref. [52]). It is not clear whether such appraisals of intensity of narrative emotion would be reflected as a more physiological (resulting in increased activation in emotion processing areas) or a more emotion-appraisal response (resulting in increased activation in orbitofrontal stimulus evaluation areas). In line with previous research on related concepts such as suspense (e.g., ref. [39]), we hypothesized that brain areas linked to attention and predictive inference are sensitive to emotional intensity. We had no specific predictions of the involvement of brain areas linked to social cognition and emotion processing since it is not conceptually or empirically clear how they relate to narrative emotional intensity.

We reanalysed a dataset from Hartung et al.[53] in which 52 participants listened to two literary short stories while neural activity was measured with functional MRI (fMRI) in a similar design as in Wallentin et al.'s 2011 study[37]. The two stories are suspenseful and gloomy literary short stories that are constructed around a protagonist and their mental world in a threatening situation. Hence, the valence of both stories is overall negative and does not shift much; so for the purpose of this study, we focus on arousal and suspense as the main components for emotional intensity. To get word-by-word ratings for appraised literariness and emotional intensity, we recruited two independent groups of participants to rate the stories used in the previous experiment in two new online surveys. These ratings were used to predict hemodynamic responses to literariness and emotional intensity in the fMRI group in a whole-brain analysis to detect brain areas that are sensitive to these two aspects of aesthetic

appreciation. We additionally correlated neural responses in regions of interest (ROIs) with measures of appreciation and individual differences in reading behaviour, social cognition, and subjective reward from solving cognitive or emotional problems. Based on these results we aimed to generate new hypotheses and questions guiding future research on neuroaesthetics of literature.

## Results

**Behavioural rating survey**. Emotional intensity and literariness ratings are visualized in Fig. 1 and Supplementary Figs. 1 and 2. The validity of these word-by-word ratings (see Supplementary Data 1 for mean word ratings and level of agreement) were verified in two steps. First, we computed their intraclass

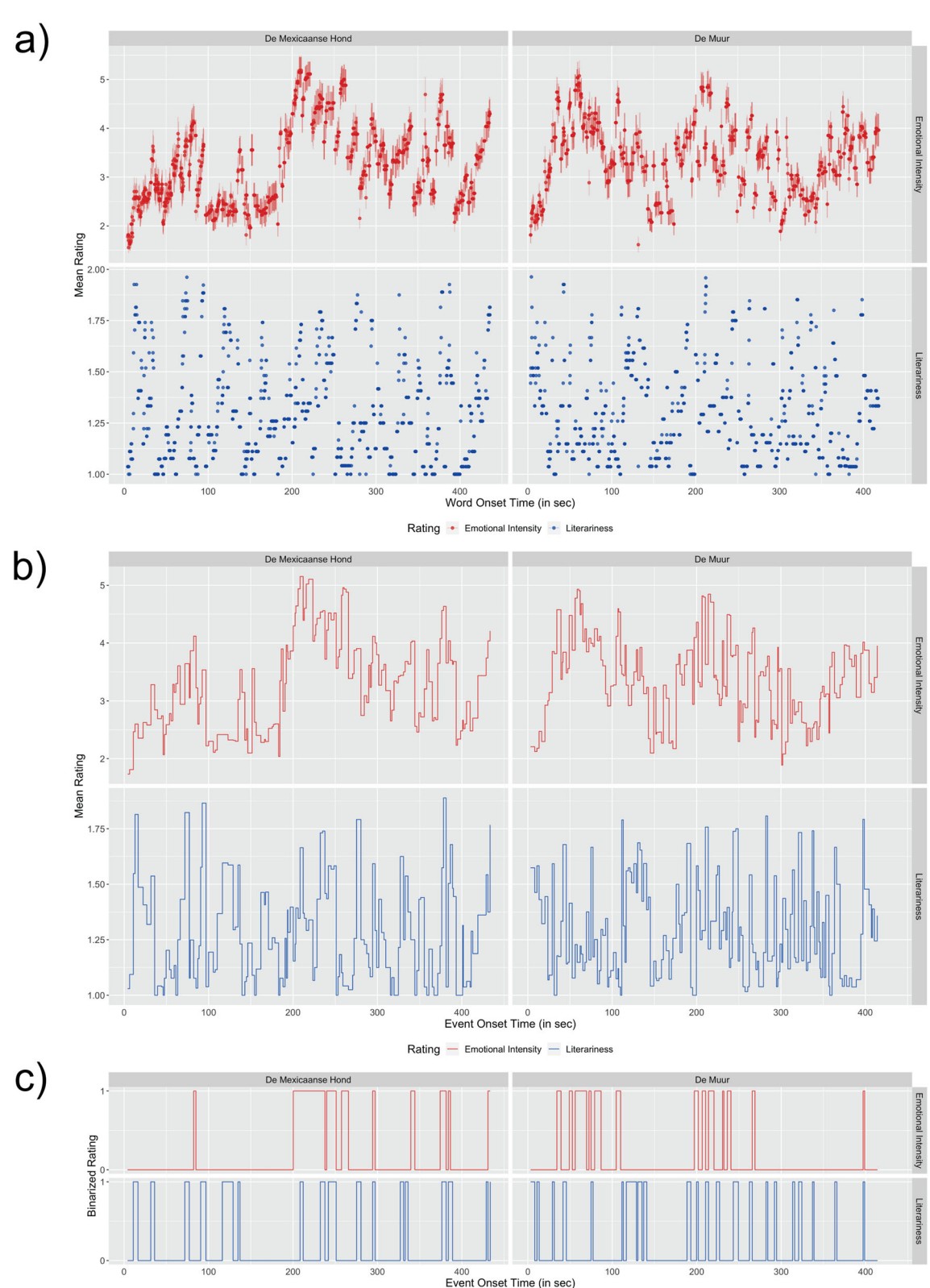

**Fig. 1 Mean ratings of emotional intensity and literariness in two stories ($N = 27$, respectively). a** Group mean ratings (with vertical bars indicating standard errors) of emotional intensity ($N = 27$, upper row, measured on a scale of 1–7, with 1 being "completely unaffected" and 7 being "felt intense emotion") and literariness ($N = 27$, lower row, a binary choice between 1 or 2, with 1 being "normal" and 2 being "stylistically remarkable/well-written") over the course of both stories per word onset time (in seconds). **b** Step plot of mean ratings of emotional intensity (on a scale of 1–7, with 1 being "completely unaffected" and 7 being "felt intense emotion") and literariness (a binary choice between 1 or 2, with 1 being "normal" and 2 being "stylistically remarkable/ /well-written") over the course of both stories against semantic event onset time. The horizontal width of each step indicates the duration of the semantic event(s) with the same rating. **c** Step plot of binarized event ratings against the semantic event onset time ("1" indicates highly emotional or literary events and "0" otherwise). The horizontal width of each step indicates the duration of the semantic event(s) with the same rating.

correlation coefficients, ICC(2,$k$), which is a measure of absolute agreement in emotional intensity or literariness ratings after accounting for systematic errors from individual raters (e.g., people may have different baseline emotional intensities, or thresholds for literariness) and random errors[54,55]. The agreement of ratings in our dataset (ICC(2,$k$)$_{emo}$ = 0.86, ICC(2,$k$)$_{lit}$ = 0.89; both values classified as "good"[54]) shows that participants reported experiencing similar emotional intensity trajectories or literary experiences. Emotional intensity ratings had relatively small standard errors and followed a continuous trend across each story, with rises at the beginning (both stories were written to quickly capture a reader's attention), in the middle (as the plot developed into major conflicts), and at the end (both stories had a surprising twist at the end) (Fig. 1a). Literariness ratings varied more between higher and lower values and highly rated segments tended to be short words or phrases which are highly salient and foregrounded in the context (Fig. 1a and see Supplementary Table 1 for segments rated the highest on literariness).

**FMRI results for literariness**
*Whole-brain analysis.* The parametric predictor with ratings of literary language correlated positively with increased activation in the left inferior and superior parietal lobules (precuneus), and left supramarginal gyrus extending into left angular gyrus. The same predictor correlated negatively with activation in bilateral superior and middle temporal gyri, and right Heschl's gyrus (see Fig. 2a and Table 1).

*ROI results*

*Correlation of aesthetic and experiential measures with Literariness ROIs.* None of the three RCs of the story rating or the two SWAS components *mental imagery during reading* and *emotional engagement with the protagonist* correlated with activation levels in ROI derived from the literariness WBA. See Supplementary Fig. 3 and Supplementary Tables 2–6 for a complete overview of results.

*Correlation of individual difference measures with Literariness ROIs.* Individual differences between participants in their reading behaviour measures correlated significantly with activation levels in several ROIs (see Fig. 3a). How much participants care about style when reading correlated negatively with activation levels in left AG ($\rho = -0.27$, $p = 0.006$) and bilateral MTG ($\rho = -0.27$, $p = 0.006$). The ART also correlated with activation in the left SPL ($\rho = 0.23$, $p = 0.02$). However, none of these correlations survive multiple comparison correction with the Bonferroni method.

None of the individual differences that were related to reward (NCS, NAS) or social cognition (EQ, IRI fantasy scale) correlated with BOLD activation in ROIs relevant for literariness.

**FMRI results for emotional intensity**
*Whole-brain analysis.* No activation clusters positively correlated with the parametric predictor of emotional intensity. The same

predictor correlated negatively with bilateral activations in post-central gyri and sulci, precunei and cunei, dorsolateral precentral gyri, mid- to posterior cingulate gyri, medial frontal gyri, as well as activations in right inferior parietal lobule, right para-hippocampal gyrus, right fusiform gyrus, right middle and inferior frontal gyri, and the anterior part of the left dorsomedial cerebellum (see Fig. 2b and Table 1).

*ROI results*

*Correlation of aesthetic and experiential measures with Emotional Intensity ROIs.* The activation levels in the FPAN ($\rho = 0.25$, $p = 0.01$) and the FCN ($\rho = 0.21$, $p = 0.03$; see Fig. 3b) were correlated with the story rating component RC1 reflecting how interesting the participant found the narrative. After multiple comparison correction (Bonferroni), only the correlation of RC1 and FPAN survived statistical thresholding. None of the experiential measures (mental imagery during reading, emotional engagement with the protagonist) correlated with activation in ROIs for emotional intensity. See Supplementary Fig. 4, and Supplementary Tables 7–11 for complete overview of results.

*Correlation of individual differences measures with Emotional Intensity ROIs.* None of the reading related individual differences such as caring about style, liking fiction, number of novels read last year, frequency of reading, and ART correlated with BOLD activity in ROIs linked to emotional intensity during reading. Individual scores on the NAS questionnaire were negatively correlated with activation in the FCN ($\rho = -0.20$, $p = 0.04$) and the FPAN $\rho = -0.22$, $p = 0.02$; see Fig. 3c). However, only the correlation between NAS and FPAN survives multiple comparison correction (Bonferroni). See Supplementary Fig. 3, and Supplementary Tables 7–11 for complete overview of results.

**Discussion**
In this study we explored the neural correlates of implicitly engaging with literary language and emotional intensity when reading narratives. Two independent groups of raters provided subjective assessments of either stylistically remarkable language or appraisals of the subjectively felt emotional intensity evoked by the text for two literary stories. These ratings were used as parametric predictors to model BOLD activity in an independent group of participants who listened to the same stories while their brain activity was measured with fMRI for comprehension and enjoyment without being asked to pay attention to emotion or literary language. We found that literariness and emotional intensity activate different neural networks providing tentative evidence for our hypothesis that different types of aesthetic responses are supported by different neural systems. Literariness was associated with brain areas linked to language processing, while emotional intensity was linked to deactivation of the frontoparietal attention and control network. Sensitivity to literary language was further correlated with reading habits. Amplitude of deactivation in the frontoparietal attention network in response to emotionally intense segments correlated with how

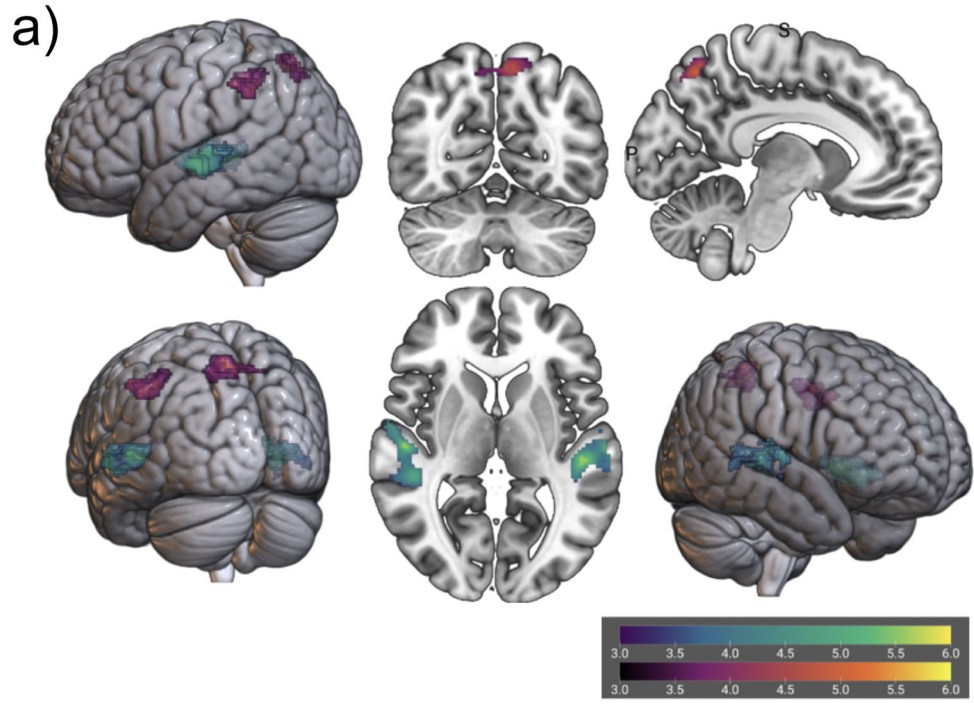

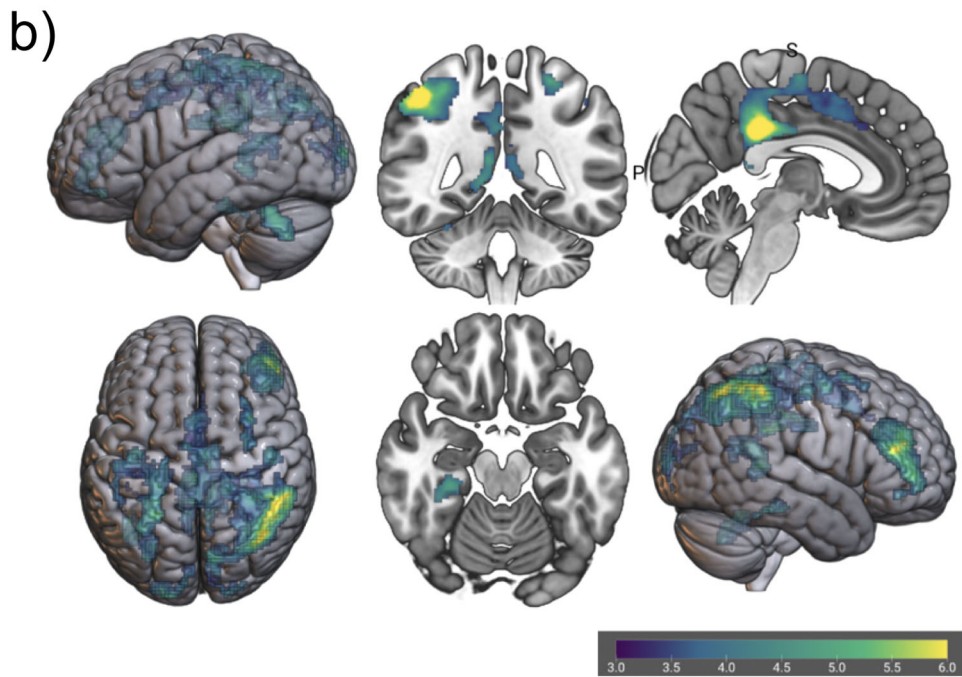

**Fig. 2 Whole-brain analysis results for literariness and emotional intensity (N = 52). a** Brain correlates of appraised literariness during narrative engagement significant at permutation-based combined cluster-voxel-extent threshold (positive correlates in purple and negative correlates in cyan). The general linear model parametrically modelled semantic events for literariness. All within-cluster voxels are family-wise error corrected at $p < 0.05$. **b** Brain correlates of appraised emotional intensity during narrative engagement significant at permutation-based combined cluster-voxel-extent threshold (only negative correlates survived thresholding, in cyan). The general linear model parametrically modelled semantic events for emotional intensity. All within-cluster voxels are family-wise error corrected at $p < 0.05$.

interesting the stories were rated. This finding shows that different aspects of aesthetic experiences are simultaneously supported by different brain systems and that aesthetic experiences cannot be reduced to a common neural basis. We discuss the findings in detail below.

**Literariness.** Segments of the stories rated as high in literariness compared to lower ratings of literariness were associated with increased activation in an area in the left temporo-parietal cortex including the angular gyrus and posterior parts of the supra-marginal gyrus, as well as a region in the left precuneus. The

**Table 1 fMRI results of whole-brain analysis for literariness and emotional intensity.**

| Contrast | | Region | Cluster size | x | y | z | Max. t value (d.f. = 51) |
|---|---|---|---|---|---|---|---|
| **Lit+** | Left | Inferior Parietal Lobule, Supramarginal Gyrus | 571 | −52 | −44 | 40 | 4.7988 |
| | Left | Precuneus, Superior Parietal Lobule | 570 | −10 | −66 | 54 | 5.2963 |
| **Lit−** | Left | Superior + Middle Temporal Gyrus, Heschl's Gyrus | 986 | −46 | −28 | 2 | 6.1302 |
| | Right | Superior + Middle Temporal Gyrus, Heschl's Gyrus | 1146 | 52 | −18 | 4 | 5.9044 |
| **Emo+** | | No suprathreshold clusters | | | | | |
| **Emo−** | — | R Inferior Parietal Lobule, L + R Mid and Posterior Cingulate Gyri, L + R Postcentral Gyri, L + R Precentral Gyri, L + R Precunei, L + R Cunei, R Middle Frontal Gyrus, L + R Medial Frontal Gyri | 13793 | 46 | −40 | 52 | 8.5226 |
| | Left | Cerebellum Posterior Lobe | 466 | −42 | −54 | −32 | 4.9969 |
| | Right | Parahippocampal Gyrus, Fusiform Gyrus | 312 | 30 | −30 | −20 | 5.104 |
| | Right | Middle + Inferior Frontal Gyrus (pars triangularis, pars orbitalis) | 1769 | 46 | 38 | 18 | 6.3787 |

fMRI group-level results using general linear model modelling semantic events with permutation-based combined cluster-voxel-extent threshold. "Lit" indicates correlation with literariness predictor and "Emo" indicates correlation with emotional intensity predictor. "+" indicates a positive correlation with the predictor and "−" indicates a negative correlation. The location of the peak t value is always reported as the first anatomical region in each cluster.

reverse contrast showed negative correlations bilaterally in middle superior and middle temporal gyri and superior temporal sulci. We did not find related activation in the left IFG with our statistical thresholding, a pattern that has previously been linked to processing of figurative language[56] (see review in ref. [57]; but see Supplementary Method, Supplementary Table 12, Supplementary Fig. 5, Supplementary Discussion, and Supplementary References for results with a more lenient thresholding that include a positive cluster in the left IFG).

Cortical areas linked to language processing, specifically semantics, are sensitive to foregrounded or literary language (rated as "stylistically remarkable/well-written"). The (bilateral) middle temporal cortex is prominently involved in semantic memory and lexical knowledge (see review in refs. [58–60]), while the left angular gyrus is hypothesized to be a core region for semantic integration[61,62] (see also ref. [59]). The left angular gyrus supposedly serves as a semantic binding hub, where meaning from multiple modality-specific regions converges to form abstract, integrated, amodal representations[61,62]. Processing literary language may require additional processing and integration of semantic meaning. An alternative explanation could be that foregrounding in literature often coincides with the expression of emotions which has been linked to saliency effects and increased workload in brain areas linked to semantic integration (unification[58]) in sentence and word processing studies (see review in ref. [24]). It is important to point out that processing of local emotions in text is different from the global intensity modulation in our emotional intensity regressor that models narrative level of emotion intensity and suspense.

Alternatively, this effect could be driven by possible lower lexical frequency of words rated as literary, low frequency of unusual constructions, or retrieval of infrequent semantic features of words[59] even though the lexical entry might be frequent. Low frequency and literariness are indistinguishable in our design since foregrounded language, by definition, occurs less frequently within a given context. One hypothesis we can derive from this observation is that low linguistic frequency correlates with experiencing language as being literary. It further opens the question if the role of the angular gyrus in comprehension is integrating rich semantic information or reacting to the increased semantic computational load. This contrast could be tested in a paradigm in which semantic complexity and strikingness are orthogonalized.

The bilateral deactivation in the middle superior temporal cortex could also be a result of the relation between perceived literariness and lower frequency in language use. These regions

(bilaterally) are involved in lexical processing and word meaning[63] and seem to causally contribute to lexical knowledge[60,64,65]. The deactivation of these regions in response to literary and hence low frequency language might reflect inhibition of default processing that facilitates semantic integration of features that are less prominent for typical word meaning. Targeted brain connectivity studies between bilateral middle temporal cortex, left angular, and left supramarginal gyrus could provide evidence for such an interaction between semantic processing regions.

Interestingly, the activation in the ROIs in the bilateral MTG and the left AG were negatively correlated with how much participants cared about style, meaning that people who prioritize the form of language show decreased activation in these areas when encountering literary language compared to participants that pay more attention to (semantic) content.

Literary language was further associated with activation in a region in the left precuneus. The precuneus is involved in many different cognitive processes including mental imagery[66], self-referential thought, episodic processing and event integration[67,68]. It is part of the extended language network where its role is thought to be to support coherence[69–71] and situation model building[71,72]. The precuneus has also been speculated to be involved in conceptual processing[73] (see review in ref. [57]), and processing of figurative language[30] but its concrete role in narrative processing is unclear. In our study, activation in this area positively correlated with individuals' scores on the author recognition test, a measure for literacy or education in reading. This means that in well-read participants this area is more sensitive to literary language. While it is hard to draw a conclusion from this finding given the various postulated roles for this region in language processing, we speculate that this region might contribute to integrating literary language in a more general contextual representation or possibly mental imagery of events which might be more practiced in educated readers. The precuneus is also part of the DMN, which has been hypothesized to play a crucial role in aesthetic appraisals[1,74]. However, since no other areas of the DMN were sensitive to literariness, we refrain from interpreting our findings as related to the DMN.

Knowledge about literary context and stylistic devices in a piece of literature can help comprehension. This aspect of aesthetic experience is linked to existing knowledge about the literature tradition such as genre, author, narrative devices, style, and historical context of a piece. Several of our measures of individual differences in reading habits and literacy correlated with sensitivity to foregrounded language in the language system.

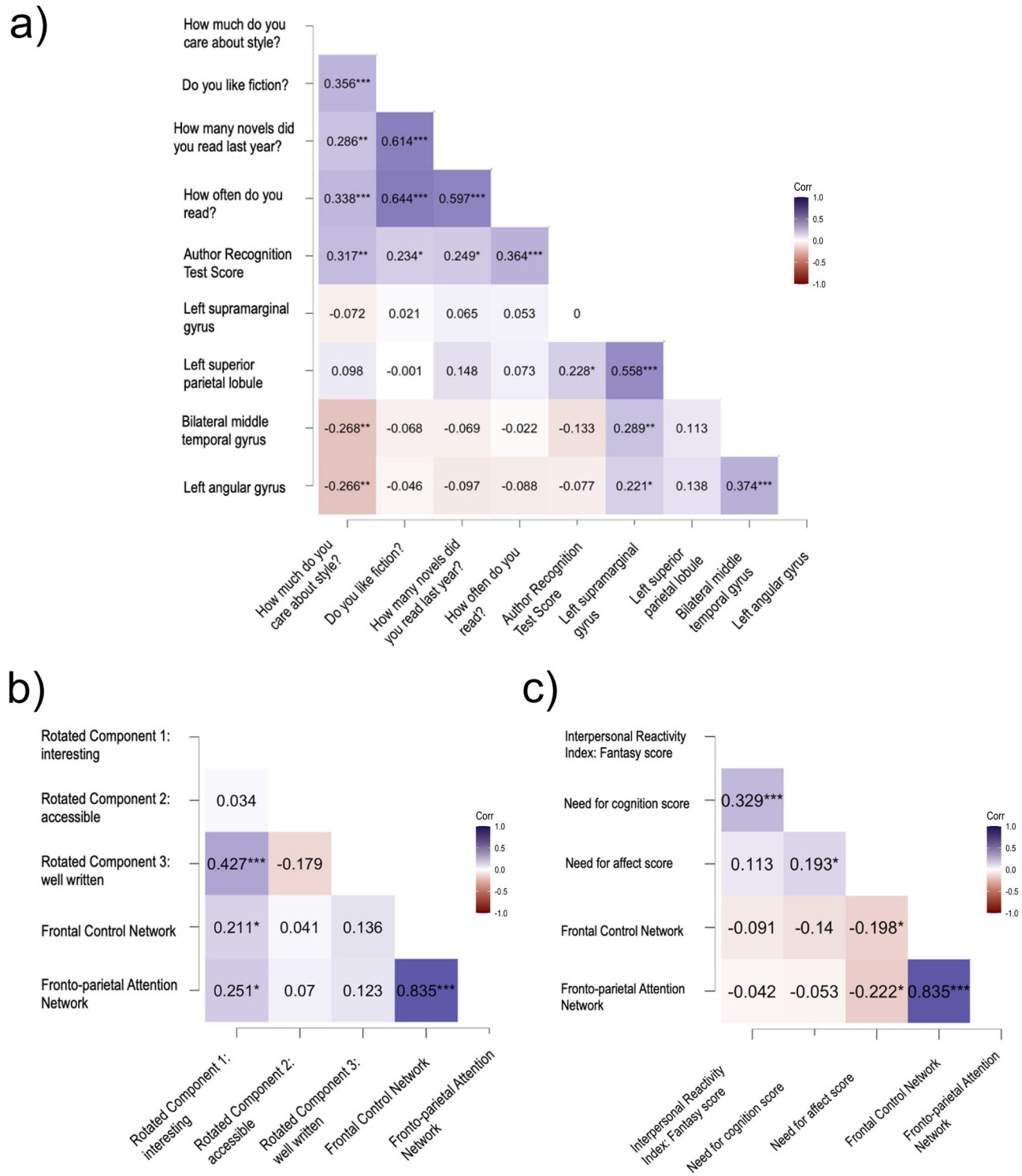

**Fig. 3 Regions of interest analysis correlating behavioural data with fMRI percent signal changes (N = 52). a** Correlation matrix with literariness-related region of interest percent signal changes and individual difference measures of reading behaviour (positive correlations in purple, negative correlations in maroon). **b** Correlation matrix with emotion-related ROI percent signal changes and principal components of story appreciation measures. **c** Correlation matrix with emotion-related ROI percent signal changes and individual differences in (social) cognition and subjective reward. *p < 0.05, **p < 0.01, ***p < 0.001.

Training effects of fiction reading likely produce more effective strategies to process literary language and impose a lower cognitive load than experienced by less trained individuals (see e.g., ref. [15]). Other than reducing cognitive load and supporting prediction in literary contexts, it is unclear how meta-knowledge and experience affect aesthetic appreciation and underlying brain systems. In the present study, we tested the effects of measures like literacy and frequency of reading fiction but do not have detailed data on individual's experience with and knowledge about literature. Large and diverse datasets could appropriately address these important individual aspects of aesthetic appreciation.

Mukařovský's theory proposed that foregrounded, literary segments "stand out" from other parts of a literary work and make readers aware of their stylistic unusualness, given preceding linguistic input. Foregrounding is contrasted with backgrounding. The latter supports immersion into a narrative (see also ref. [19]). In our dataset this claim is confirmed by the fact that ratings for perceived emotional intensity and literariness from independent groups of raters are almost mutually exclusive. Foregrounding materializes at all levels of language processing (e.g., semantic, syntactic, or phonological) and as statistical outliers of linguistic features they are more salient. Increased brain activations with literary segments hence may reflect attentional modulation of the language network[75]. Literary segments appear to modulate language processing because statistically improbable linguistic features likely require greater integration of meaning and the inhibition of default processing. This suggestion is supported by the fact that activation in these brain areas were linked to individual differences in familiarity with literary fiction but not to measures of aesthetic or experiential responses in narrative engagement (cf. ref. [19]).

**Emotional intensity**. This part of the study was largely exploratory as we had no specific hypotheses with respect to predictions of neural activity reflecting the appraisal of emotional intensity. No brain areas showed increased activations associated with high emotional intensity at the statistical threshold we applied (see Supplementary Method, Supplementary Table 12, Supplementary Fig. 6, Supplementary Discussion, and Supplementary References for results with a more lenient thresholding). However, judgments of increased emotional intensity were linked to deactivation in a large set of areas including bilateral postcentral gyri and sulci, precunei and cunei, posterior and middle cingulate cortex, medial frontal gyrus, as well as in right inferior parietal lobule, right IFG, and middle frontal gyrus. This set of areas resembles the frontoparietal network for controlled attention (FPAN) and the frontal control network (FCN). The FPAN has been implicated in tasks related to (top down) attention and executive control[76–79]. The observation that greater emotional intensity during engagement with literature is linked to decreased activation in the FPAN might be linked to suspended executive control and directed attention, as participants' attention is guided by the story, rather than their active seeking of information. Similar findings have been shown for the related concept of narrative suspense[39,49].

One core area of the FPAN that shows one of the largest effects in our study, the PCC, has been implicated as a central hub that controls internally vs externally directed attention and cognition[80,81]. We hypothesize that this deactivation may be linked to self-relevance in engaging with literature or narratives in general.

Another area that showed a large deactivation in response to intense emotional segments during literature engagement is a region in the right inferior parietal cortex extending into the temporo-parietal junction including the right supramarginal, angular, and inferior parietal gyri. This cluster overlaps with areas often associated not only with the FPAN but also with the mentalizing network. It is possible that during intense moments in a narrative, recipients pay less attention to the mental world of fictional characters and instead embrace their own emotional experience. This interpretation is in line with the simultaneous large deactivation in right IFG, an area which seems to be important for narrative understanding and event coordination as part of the extended language network[82]. We hypothesize that during emotionally intense moments, readers focus less on the

events and characters in a narrative, and more on their own experience.

The results from the ROI analysis with individual differences further support the interpretation that suspense of controlled attention is an important aspect of getting lost in a narrative during intense moments. The deactivation of both the FCN and FPAN ROIs was linked to individual differences in need for affect (NAS) scores—a measure of reward experienced when engaging with social and emotional situations. The higher an individual scored on the NAS, the stronger the deactivation of the FCN and FPAN during emotional intense moments. Furthermore, the activation of both the FCN and FPAN ROIs was also correlated with how interesting participants rated a story to be.

Why did we not find any positive correlations in brain areas linked to emotional intensity? We think this observation might be attributed to the fact that the emotional intensity ratings were not from the same participants as from whom we gathered the functional MRI data. While the group rating data on emotional intensity is highly correlated across all subjects in the rater group, appraisals of emotional intensity are subjective. Future studies could implement experimental designs that allow exploration of individual variability of different aesthetic experiences with physiological and neuroimaging data.

It might also be surprising that we did not find activation in emotion processing areas given the highly negatively valenced stories we used and that previous research has reported activation in primary emotion areas such as amygdala (e.g., refs. [35,37]). One difference between our study and the studies that report such effects is that our narratives were completely new material to the participants. Hsu et al.[35] recruited specifically participants who were familiar with the Harry Potter storyverse to test for their emotional responses during engagement with passages that were already known to the participant. Similarly, Wallentin et al.[37] used an extremely well-known folk tale *The Ugly Duckling* that is so enculturated that every participant could be assumed to have known this story since early childhood. It is hard to speculate about the potential differences between an individual and a cultural or communal experience of story engagement but it is clear that plot level prediction violation is substantially less likely for highly familiar stories and that the emotion processing is affected from memory of previous experiences of the story. The dissociation of emotional responses to novel vs familiar and individual vs enculturated stories is an extremely interesting and important avenue to pursue for future research.

Overall, highly emotional contents in our study appear to disengage FPAN and hence controlled attention and top down information seeking. We speculate that participants temporarily suspend control of directed attention and allow the stories to guide their cognition during emotional segments. Further research is needed to test whether deactivation in control and attention networks predicts different aspects of emotional engagement with stories and aesthetic experiences linked to affect or emotion regulation. Moreover, the connection with individual differences in reward to different types of content suggests that readers differ in their motivation to read and enjoy different types of reading materials.

**General discussion**. Our results support the hypothesis that literary form and fluctuation in emotional intensity during narrative engagement are facilitated by distinct neural systems which suggests that different aspects of aesthetic engagement rely on different processing systems. Brain areas associated with semantic processing seem to be sensitive to stylistic features while suspense of attention and executive function may mediate emotional

content responses in aesthetic appreciation. These aspects of aesthetic appreciation can be related to the knowledge-meaning system (literariness) and the emotion-valuation system (emotion intensity or suspense) of the aesthetic triad framework[83]. Interestingly, we found that narrative induced emotions (such as suspense) are linked to aesthetic experiences during literary engagement, while aesthetic responses to the writing of a piece of literature were linked to individual differences in experience with reading literary texts and awareness of style.

Few studies have investigated continuous aesthetic experiences as a narrative unfolds (but see ref. [37] for a similar study in which they asked participants to rate their emotional experiences line by line on paper; see ref. [39] for an interspersed reading-rating approach). Our online survey enabled participants to rate every word differently without imposing boundaries. This high granularity of rating may be especially important for judging literariness. As shown in Fig. 1a, people tended to rate specific words rather than sentences as literary, and this information allowed us to model haemodynamic responses at a finer time scale.

Aesthetic appreciation generally activates areas linked to reward and pleasure, yet we did not find related regions such as the striatum in our analysis. While the current study deconstructed aesthetic appreciation into temporal ratings of two aspects of aesthetic engagement with narrative, there are many other aspects of aesthetic engagement and more importantly, it is difficult experimentally to anchor the experience of pleasure to specific time points. This methodological challenge for experimental designs needs to be addressed in future directions of neuroaesthetics to advance our understanding of the relation between aesthetic experiences—as we measured here—and aesthetic evaluation and the neural correlates of such an interaction.

A much discussed possible neural correlate for aesthetic appreciation is the DMN which is hypothesized to be involved in aspects of self-referential aesthetic experiences[1,74]. Associations between the DMN and aesthetic experiences are based on fMRI evidence from aesthetic experiences in visual art[84]. Whether these findings generalize to other types of aesthetic experiences is unclear but in our study, we find no indication that the DMN was linked to aesthetic experiences when people engage with literary stories. Given the many cognitive processes that engage the DMN, we are cautious about implicating the DMN in all aesthetic processes.

Our current study has limitations. Conceptually, we need further clarification on the ontology of "emotional intensity" and "literariness" as semantic concepts and as deconstructable phenomena that can be manipulated for neuroimaging studies. The values used in parametric modulations of emotion and literariness were averaged ratings from separate groups, which might not reflect individual fMRI participant's aesthetic experiences. High granularity ratings from the same participants whose brain activity is repeatedly measured with different literary materials could offer a more accurate understanding of the meaning of these neural correlates.

One crucial issue is the distinction between appraisals of emotion and physiological emotional state (see ref. [52] and references therein). This distinction may explain the absence of subcortical activations typically associated with emotion (e.g., see review in ref. [85]). Experimentally, the dominant changes in emotional intensity from a story arc develop over several minutes, an effect that may not be easily detectable because of the slow signal drift noises in fMRI. Future experiments could consider using literary stories of varying length and adjust high pass filters to optimize signal to noise ratio. More diverse stimuli in terms of topic, valence, and style could also be used to understand their

shared neural responses as well as variabilities in stories and individuals.

Our reverse inferences of correlational activations need to be tested prospectively. Individual variability in brain regions and their connectivity profile, further divisions in each anatomical area that are beyond the detection ability of fMRI[86] (see ref. [87] for an example illustrating subdivisions in angular gyrus), and the nature of reverse inference[88] (also see ref. [89]) are general problems that are relevant to the current study. Possible future directions include multiple session studies with fewer participants but more varied story materials, and careful meta-analyses that combine both forward and reverse inference for stronger function-to-area and function-to-network mapping[90,91], particularly for large networks like the extended language network or the FPAN. Dynamic network modelling to dissociate interacting networks as well as nodes within each network (whether they are specific to a function or are domain specific such as attention modulation) may be helpful as well[86,92].

Aesthetic emotions—in literature engagement and beyond—remain a controversial and elusive concept[1–5]. During engagement with literature, sensory, cognitive, attentional, and emotional experiences dynamically interact with and modulate one another. The aesthetic triad framework[83] proposes that aesthetic experiences result from a dynamic interaction between sensory-motor, emotion-valuation, and knowledge-meaning neural systems. We confirm that the distinction between form and content in the visual arts also applies to literature. Brains simultaneously process stylistic form and emotional content of literature through different neural structures and the quality of activations in these neural systems is linked to aesthetic experiences and training with a certain art form (here literary narratives). We confirm that literariness is instantiated in language-specific neural systems associated with semantic integration. It appears that domain-specific subsystems (e.g., perceptual, language, and motor systems) facilitate appraisal of stylistic features that contribute to literary aesthetic experiences whereas domain general attention and executive control systems interact and modulate emotional appraisal. Based on our finding that experiencing emotional intensity during engagement with literature is linked to deactivation of controlled attention networks, we propose that appraisal of intense emotional content releases the reader from executive control.

## Methods

This study combined two datasets with independent groups of subjects. Online behavioural data were collected from two groups (each $N = 27$) of participants who rated two stories word-wise for either literariness or emotional intensity. These ratings were used to model BOLD response in a previously collected fMRI dataset ($N = 52$) of participants who listened to the same two stories. There was no overlap in participants between any of the groups. From each group we also collected ratings of enjoyment and several individual differences measures querying reading behaviour and attitudes (see Table 3 and Supplementary Table 13 for comparison of the groups).

**Stories**. In Hartung et al., 2017[53], two Dutch literary short stories were recorded by a native speaker in Dutch. De Muur ("The Wall," DM) by Peter Minten (published in 2013, 1121 words) and De Mexicaanse Hond ("The Mexican Dog," DH) by Marga Minco (published in 1990, 1236 words) are both typical short fiction stories describing a single incident in the respective protagonist's life. The stories are written from the protagonist's perspective. Both original stories used first-person pronouns to refer to the protagonist. For purposes of the original study (see Hartung et al.[53]), the stories were also recorded with third-person pronouns referring to the protagonist. The changes were minimal as only pronouns and dependent verb conjugation were affected by this manipulation (113 word changes in DH; 93 word changes in DM; see Supplementary Note for English translations). The recordings were about 7 min for each story. We used audio recordings instead of text presentation to assure identical timings and durations in stimulus presentation across all subjects in the experimental design.

In the current study, we re-used the audio recording to acquire additional ratings to keep the recent rating studies as close to the original experimental design as possible.

### Aesthetic and experiential measures of engagement with stories

*Story appreciation.* In Hartung et al., 2017[53] and the current study, story appreciation was measured directly after listening with ten adjectives that correspond to different dimensions of appreciation (translated into Dutch from ref. [93]). The list consisted of the following items: interesting, well-written, of high literary quality, easy to understand, accessible, thrilling, beautiful, fascinating, emotional, and sad. To reduce the dimensionality of the data we performed a principal component analysis (PCA) on these ten items. This PCA was performed on the data from the two rater groups and then components were correlated with BOLD activity in the fMRI group. The PCA was based on eigenvalues (>1) with oblique rotation of factors (promax) and component decomposition based on the correlation matrix. The PCA resulted in three rotated components (RC): interest in the story (RC 1

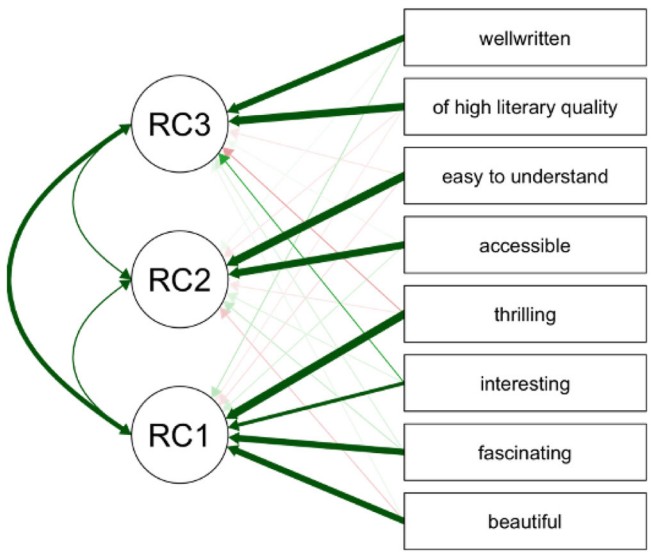

**Fig. 4 Principal component analysis on ten-item story appreciation measures from the behavioural rater group (N = 54).** Two items ("emotional" and "sad") were excluded due to data collection errors. The remaining eight items resulted in three rotated components (RC) with one linked to interest in the story (RC 1 "interesting"), one linked to ease of understanding and accessibility (RC 2 "accessible"), and one component linked to quality of writing (RC3 "well-written").

"interesting"; items: interesting, thrilling, beautiful, fascinating), ease of understanding (RC 2 "accessible"; items: easy to understand, accessible), and quality of writing (RC3 "well-written"; items: well-written, of high literary quality, see Fig. 4). Because of a data collection error in the current study, the items emotional and sad were not collected for the online survey rating and hence not included in the PCA and subsequent analyses.

*Experiential aspects of reading.* In Hartung et al., 2017[53] and the current study, two subscales of the story world absorption scale (SWAS,[94]) were measured after each story. This measure consisted of 15 items testing (a) emotional engagement with the protagonists (9 items), and (b) the experience of mental imagery (6 items; see Table 3 for translated items). In addition, there was one item addressing third perspective taking (first-person perspective taking is already addressed by one item in the mental imagery scale) as part of the original study's research question, which was not used for the current study.

**Individual differences.** In Hartung et al., 2017[53] and the current study, for all three participant groups, we collected measures for reading habits. We additionally collected several measures for individual differences from the participants in the fMRI sample including the fantasy scale of the Interpersonal Reactivity Index (IRI[95]), Need for Cognition (NCS[96]), Need for Affect (NAS[97]), Author Recognition Test (ART[98,99]), and the Empathy Quotient (EQ[100]) questionnaire in this order (see Table 2 for overview).

*Reading habits.* In Hartung et al., 2017[53] and the current study, to assess participant's self-reported reading habits, we used three items assessing liking of fiction ("Do you like fiction?"), frequency of reading ("How often do you read?"), and the number of books read in the past year ("How many novels/stories did you read in the past year?"). The answer choices of liking of fiction were on a scale from 1 (not at all) to 7 (totally) in the fMRI sample, and from 0 (not at all) to 100 (totally) in the behavioural rater samples (which was converted to the 7-point Likert scale to match the fMRI sample).

The fMRI sample in Hartung et al., 2017[53] was additionally assessed on their sensitivity to style ("How much attention do you pay to style when reading?") on a scale from 1 (not at all) to 7 (totally). For the frequency of reading question participants chose between daily, more than twice per week, once per week, not regularly, and never. The question regarding the number of books was a numerical estimate.

*Additional measures in fMRI sample.* In Hartung et al., 2017[53], for an estimate of print exposure, we used a Dutch version of the ART. ART contains 42 names, of which 30 are actual fiction authors and 12 are made up names. Participants mark the names they recognize. The score of each participant is computed by subtracting the sum of all incorrect answers from the sum of all correct answers. The total score can vary between −12 (only non-existent author names selected) to 30 (all correct names selected).

We used the six items from the Fantasy scale of the Interpersonal Reactivity Index (IRI). IRI is a self-report measure of individual differences in social sensitivity, consisting of four subscales. The Fantasy scale of the IRI tests individual readiness to get transported imaginatively into the feelings and actions of fictive characters in narratives. We also included the Empathy Quotient questionnaire to

## Table 2 Overview of behavioural questionnaires.

| | Behavioural rater groups (N = 54) | | FMRI group (N = 52) |
|---|---|---|---|
| | Rated for literariness (N = 27) | Rated for emotional intensity (N = 27) | |
| Story-specific measures | 10-Adjective Story Appreciation (reduced to 3 components with principal component analysis) Experiential measures: a. Emotional engagement with the protagonist b. Mental imagery | | 10-Adjective Story Appreciation (reduced to 3 components with principal component analysis) Experiential measures: a. Emotional engagement with the protagonist b. Mental imagery |
| Individual difference measures | Reading habits (3 items): a. Liking of fiction b. Frequency of reading c. Number of novels read last year | | Reading habits (4 items): a. Liking of fiction b. Frequency of reading c. Number of novels read last year d. Sensitivity to style Author Recognition Test (ART) Fantasy Scale of the Interpersonal Reactivity Index (IRI) Need for Cognition (NCS) Need for Affect (NAS) Empathy Quotient (EQ) |

Overview of story-specific and individual difference measures administered to the independent behavioural rater groups and functional MRI group.

**Table 3 Questionnaire items and results overview.**

| | | De Mexicaanse Hond | | | | | | De Muur | | | | | |
|---|---|---|---|---|---|---|---|---|---|---|---|---|---|
| | | fMRI group | | | Rater group | | | fMRI group | | | Rater group | | |
| Questionnaire name (story-specific) | Question (on a scale of 1–4 with 1 being "completely disagree" and 4 being "completely agree") | Mean | SD | Cronbach's alpha | Mean | SD | Cronbach's alpha | Mean | SD | Cronbach's alpha | Mean | SD | Cronbach's alpha |
| Appreciation Questionnaire | Interesting | 2.04 | 0.94 | NA | 2.63 | 0.98 | NA | 1.73 | 0.95 | NA | 3.04 | 0.85 | NA |
| | Well-written | 2.06 | 0.93 | | 3.00 | 0.80 | | 1.73 | 0.77 | | 3.20 | 0.66 | |
| | Of high literary quality | 2.24 | 0.99 | | 2.94 | 0.79 | | 2.21 | 0.89 | | 3.04 | 0.78 | |
| | Easy to understand | 2.29 | 0.92 | | 3.33 | 0.87 | | 1.88 | 1.02 | | 3.26 | 0.81 | |
| | Accessible | 2.41 | 0.92 | | 3.17 | 0.88 | | 2.27 | 1.01 | | 3.07 | 0.77 | |
| | Thrilling | 2.33 | 1.01 | | 2.80 | 1.05 | | 1.96 | 0.97 | | 3.15 | 0.98 | |
| | Beautiful | 2.75 | 0.84 | | 2.31 | 0.99 | | 2.35 | 0.93 | | 2.59 | 0.88 | |
| | Fascinating | 2.16 | 0.92 | | 2.67 | 1.05 | | 2.04 | 1.01 | | 2.85 | 0.90 | |
| | Emotional | 2.78 | 0.94 | | * | * | | 2.37 | 1.03 | | * | * | |
| | Sad | 2.96 | 0.92 | | * | * | | 3.29 | 0.91 | | * | | |
| SWAS (Imagery) | Sometimes I had the feeling that I could see through the eyes of the main character. | 1.96 | 0.88 | 0.4 | 3.00 | 0.89 | 0.87 | 1.96 | 0.99 | 0.75 | 3.17 | 0.80 | 0.81 |
| | While reading I saw situations that were described to me as if I was there myself. | 1.75 | 0.68 | | 3.02 | 0.94 | | 1.75 | 0.81 | | 3.15 | 0.96 | |
| | Sometimes I had the feeling that I was in the environment in which the story took place. | 2.12 | 0.92 | | 2.85 | 1.04 | | 2.04 | 0.99 | | 3.02 | 0.90 | |
| | Sometimes I saw the environment in which the story takes place in front of me. | 1.75 | 0.65 | | 3.28 | 0.90 | | 1.52 | 0.70 | | 3.39 | 0.76 | |
| | While reading, I saw situations that were described to me as if I were a silent spectator. | 1.98 | 0.87 | | 2.74 | 1.01 | | 1.94 | 0.87 | | 2.87 | 1.13 | |
| | While reading this story I saw a picture of the main character in front of me. | 2.67 | 1.17 | | 2.30 | 1.00 | | 2.19 | 1.05 | | 2.72 | 1.07 | |
| SWAS (Emotional Engagement) | I felt like the main character felt. | 2.79 | 1.04 | 0.83 | 2.37 | 1.00 | 0.96 | 2.96 | 1.05 | 0.85 | 2.17 | 0.97 | 0.93 |
| | I could empathize with the characters. | 2.21 | 0.96 | | 2.76 | 0.93 | | 2.29 | 1.02 | | 2.54 | 1.02 | |
| | I sympathized with the main character. | 2.13 | 0.89 | | 2.63 | 0.98 | | 2.10 | 0.93 | | 2.96 | 0.89 | |
| | I shared the protagonist's emotions. | 2.63 | 0.93 | | 2.43 | 1.00 | | 2.87 | 0.89 | | 2.52 | 1.06 | |
| | I felt sorry for the protagonist. | 2.48 | 1.04 | | 2.48 | 1.06 | | 2.13 | 1.10 | | 3.06 | 0.94 | |
| | As I read, I could imagine what it would feel like to be in the protagonist's shoes. | 2.27 | 0.95 | | 2.85 | 1.07 | | 2.40 | 1.09 | | 2.70 | 1.04 | |
| | I knew exactly which emotions the characters experienced. | 2.77 | 0.94 | | 2.41 | 1.00 | | 2.77 | 0.96 | | 2.46 | 0.97 | |
| | I could imagine how the main character felt. | 2.08 | 0.84 | | 2.85 | 0.98 | | 2.12 | 0.92 | | 2.94 | 0.94 | |
| | I understood how the main character felt. | 2.33 | 0.90 | | 2.80 | 0.98 | | 2.31 | 0.98 | | 2.67 | 1.08 | |
| Reading habits (person-specific) | How much do you like fiction? (1–7 Likert scale with 7 being "I love fiction") | fMRI group Mean 5.19 | SD 1.48 | | Rater group Mean 5.28 | SD 1.58 | | | | | | | |
| | How many novels did you read last year? | 7.64 | 12.01 | | 6.52 | 6.45 | | | | | | | |

Behavioural measures of fMRI and both rater groups assessing their overall engagement with DH and DM and individual reading habits. The rater group originally answered these questions on a scale of 0—100, and the results were converted to be on a scale of 1–4 (for story-specific questionnaires) and a scale of 1–7 (for reading habits) to match with the results from the fMRI group. Due to data collection errors, ratings of "emotional" and "sad" from the rater group were excluded (marked by "*").

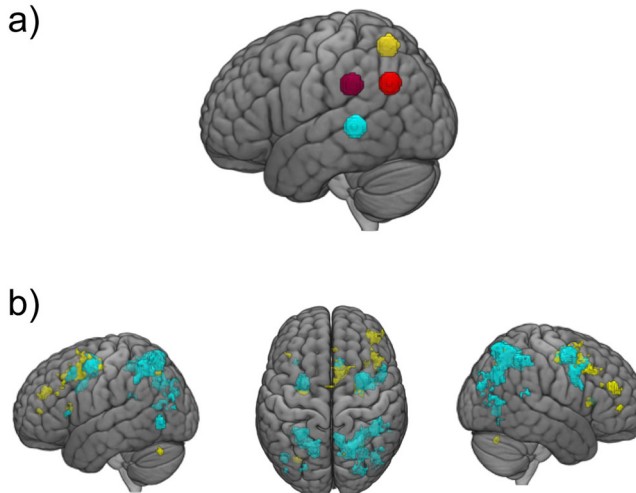

**Fig. 5 Regions of interest for literariness and emotional intensity. a** Regions of interest based on the Literariness predictor in the left supramarginal gyrus (burgundy), left angular gyrus (red), left superior parietal lobule (precuneus, yellow), and bilaterally a region stretching from the middle to the superior temporal gyrus (cyan, right location not shown, bilateral regions of interest treated as one region for the purpose of this analysis). **b** Regions of interest based on the Emotional Intensity predictor include the frontal control network (FCN, yellow) and the frontoparietal attention network (FPAN, cyan).

measure individual differences in empathy, (standardized Dutch version http://www.autismresearchcentre.com/arc_tests).

In addition, we used the Need for Cognition Scale to measure motivation to solve complex tasks, and the Need for Affect Scale to assess motivation to approach or avoid emotions. See Supplementary Fig. 7 and Supplementary Data 2 for correlations between individual difference measures with each other.

**Behavioural rating survey**. In the current study, the behavioural rating survey was conducted under approval of the local ethics committee of the Centre for Language Studies at Radboud University (number 8976). All participants gave informed consent before participating in the survey. Two independent groups of native Dutch speakers from Radboud University Nijmegen were recruited to rate the stories through a Qualtrics survey (Qualtrics, Provo, UT). In the survey, each story was presented to participants auditorily in segments of 100−200 words (9 segments for DH and 8 segments for DM, segmented at paragraph breaks, see Supplementary Note) followed immediately by its transcript on a separate page to rate before the consecutive auditory segment was presented. Participants were instructed to rate based on their experience of listening rather than reading the segment. The rating process allowed participants to select and rate text flexibly (words, phrases, sentences, or paragraphs). Participants rated all words in the story. One group ($N = 27$, 22 female, age range: 18−26 years, M = 20.8, SD = 2.7) rated the stories for literariness (forced binary choice between "stylistically remarkable/well-written" and "normal"). The other group ($N = 27$, 23 female, age range: 18−27 years, M = 21.4, SD = 2.6) rated both stories for their subjective level of emotional intensity (on a scale of 1 ("completely unaffected") to 7 ("felt intense emotion")). Within each rater group, the order of the two stories was counterbalanced. Only recordings and transcripts of the original stories, which were in first-person perspective, were presented.

To ensure adequate quality of data, participants answered a multiple-choice question about the content of the audio segment (four choices, one correct answer) after each audio Qualtrics survey page. If the question was answered incorrectly, the participants moved directly to the next audio segment without rating the corresponding transcript of the audio segment (24 out of 918 segments (<3%) were skipped because of incorrect answers). Data from all participants were included in the analysis.

After each story, participants completed the story-specific measures, including story appreciation and experiential measures of story engagement and mental imagery. Because of data collection errors, ratings for "sad" and "emotional" in the ten-item story appreciation measure were excluded. For all story-specific measures, behavioural rating participants gave responses on point sliders ranging from 0 to 100 (0 = completely disagree, 100 = completely agree). The 0–100 ratings were converted afterwards to match the 4-point scale used in the fMRI study (i.e., 0–25 is converted to 1; 26–50 converted to 2; 51–75 converted to 3; 76–100 converted to 4). This conversion ensures that the standard deviations between fMRI and rater

groups can be compared; see Supplementary Table 13 for Bonferroni-corrected t test results. At the end of the survey, participants completed the individual difference measures about their reading habits. All survey participants were compensated with either university credits or gift cards.

Both appreciation questionnaire and experiential measures (SWAS subscales) are established measures of reading engagement and used in previous studies (e.g., refs. [53,94]). To test for consistency of story-specific questionnaire measures between the behavioural rater groups and fMRI group, we calculated the mean and standard deviation of each item and carried out Bonferroni-corrected t tests (see Table 3 and Supplementary Table 13 for results). The behavioural rater groups gave statistically significant higher ratings in the appreciation questionnaire than the fMRI group did, indicating greater narrative enjoyment across different dimensions among behavioural raters. This finding was likely due to the difference in comfort of their reading situation: when listening to the stories, the fMRI participants lay in a small, dark, and noisy scanner and had to remain still, while behavioural raters likely completed the survey in the comfort of their home or library, and could take breaks in between.

We additionally calculated Cronbach's alphas (coefficient alphas,[101,102]) to test for internal consistency in experiential measures of mental imagery and emotional engagement with the protagonist for each story and for the fMRI group and both rater groups (see Table 3). Other than mental imagery item ratings by the fMRI group, internal consistency for SWAS subscales were good (>0.8).

**FMRI**. The neuroimaging data were reanalysed from Hartung et al.'s (2017)[53] study in which a group of native Dutch speakers ($N = 52$, 29 female, age range: 18–35 years, M = 23.1, SD = 3.4, 8 left-handed) listened to the two stories while undergoing fMRI. There was no overlap between the three groups of participants. In the original experiment, participants listened to one of the stories with first-person pronouns and the other story with third-person pronouns referring to the protagonist. The order of stories and pronoun condition was pseudorandomized with equal proportions. Participants were instructed to listen to the stories for enjoyment.

After each story, participants responded to the ten-item story rating and experiential measures with a four-button response device using their right hand (index finger = disagree (1), little finger = agree (4); numbers manually reversed for one participant who responded with the left hand because of a hand injury). After the scanning session, participants completed a post-scan test battery of the individual difference measures discussed above.

*Data acquisition and pre-processing*. In Hartung et al., 2017[53], images of blood-oxygen-level-dependent (BOLD) brain activity were acquired with a 3T Siemens Magnetom Trio scanner (Erlangen, Germany) with a 32-channel head coil. Cushions and tape were used to minimize head movement. Functional images were acquired using a fast T2-weighted 3D EPI sequence[103], with high temporal resolution (TR: 880 ms, TE: 28 ms, flip angle: 14°, voxel size: 3.5 × 3.5 × 3.5 mm, 36 slices). High resolution (1 × 1 × 1.25 mm) structural images were acquired using an MP-RAGE T1 GRAPPA sequence. Data were pre-processed using the Matlab toolbox SPM8 (http://www.fil.ion.ucl.ac.uk/spm). Images were motion-corrected and registered to the first image of each scanning block. The mean of the motion-corrected images was co-registered with the individual participants' anatomical scan (mean of functional images for two participants in which the T1 scan was missing). The anatomical and functional scans were spatially normalized to the standard MNI template. Finally, all data were spatially smoothed using an isotropic 8 mm full width at half maximum (FWHM) Gaussian kernel.

**Statistics and reproducibility**
*Behavioural rating survey*. In the current study, the average ratings for literariness and emotional intensity were matched with the onset time and duration of every word in both stories (Fig. 1a). We then averaged the ratings within each semantic event in the story (mostly spanning between 1 and 3 s) to obtain the mean level of emotional intensity and literariness (Fig. 1b). A semantic event was defined as a minimal segment of one or more linguistic phrases that allow lexical meanings to be integrated into a single event. For example, "as soon as Mr. Kuisters saw me coming in, /he pushed the sliding door between the store and the living room open /and said that I should go inside, /because it would take a while" would be separated into four events. To ensure that the parametric predictors are orthogonal, we tested for correlation between literariness and emotional intensity ratings in both stories. There was minimal correlation between these predictors ($p = 0.00466$, Pearson's $r = 0.157$). The coefficient of determination ($R^2 = 0.0246$), which is the square of Pearson's $r$, means that <2.5% of variance in one predictor could be accounted for by the other.

*FMRI whole-brain analysis*. In the current study, statistical analyses were performed using the Matlab toolbox SPM12 (http://www.fil.ion.ucl.ac.uk/spm) on the single-subject level. Separate general linear models (GLMs) were used to model emotional intensity and literariness with the two stories as separate sessions. The beta weight for each predictor was estimated for the time course of each voxel, using multiple regression analysis[104]. Within each GLM, we used an event-related

**Table 4 Post-hoc regions of interest selection.**

| Label | Neurosynth term | x | y | z | Pick cluster | Cluster extent threshold | z-score threshold | ROI from | ROI radius (mm) | # Studies | # Coordinates |
|---|---|---|---|---|---|---|---|---|---|---|---|
| Left superior parietal lobule | Superior parietal | −21.0 | −60.3 | 58.4 | Y | NA | 8 | Lit+ | 8 | 622 | 24,152 |
| Left supramarginal gyrus | Supramarginal gyrus | −57.3 | −38.5 | 32.4 | Y | NA | 4 | Lit+ | 8 | 296 | 11,421 |
| Left angular gyrus | Angular gyrus | −48.3 | −64.6 | 33.1 | Y | NA | 6 | Lit+ | 8 | 310 | 10,782 |
| Left middle temporal gyrus | Comprehension | −56.4 | −41.2 | 2.7 | Y | NA | 8 | Lit− | 8 | 424 | 15,365 |
| Right middle temporal gyrus | Comprehension | 54.3 | −29.4 | −1.5 | Y | NA | 8 | Lit− | 8 | 424 | 15,365 |
| Frontal control network (FCN) | Control | NA | NA | NA | N | 10 | 0 | Emo− | NA | 3796 | 137,024 |
| Fronto-parietal attention network (FPAN) | Attention | NA | NA | NA | N | 10 | 0 | Emo− | NA | 1831 | 65,346 |

Post-hoc regions of interest (ROIs) built from association test maps in Neurosynth, based on results from the whole-brain analysis with the smaller cluster-extent threshold. "Pick Cluster" indicates if a single cluster was picked to generate centre-of-mass coordinates for each spherical ROI. "ROI from" shows whether a similar region presented positive or negative correlations with emotion or literariness parametric modulation in the whole-brain analysis. Percent signal changes of an isolated 2 s event was calculated in each ROI from the single-subject level general linear model with the appropriate emotion or literariness event predictor. "# Studies" and "# Coordinates" show the number of studies and coordinates parsed by Neurosynth at the time of retrieval.

design that modelled semantic events parametrically by their mean ratings of either emotional intensity or literariness. For sessions in which participants listened to stories with third-person pronouns, the mean event ratings were calculated without the mismatched words. Six motion regressors of no interest and one constant predictor to account for difference in mean session signal were added and orthogonalized to the two predictors. Only the event predictor and its parametric modulation (rating values) were convolved with the canonical hemodynamic response function, and the predictor of parametric modulation was orthogonalized to the event predictor. A high-pass filter was then applied to minimize low-frequency noise. To ensure that the signals of interest were not filtered out, we determined the high-pass filter size separately for emotional intensity and literariness by visual inspection of the power spectrum of each parametric predictor in frequency space after a fast Fourier transform. The default high-pass filter size of 128 s in SPM was used for literariness while a filter size of 256 s was used for emotional intensity so that the prominent emotional fluctuations throughout the 7-min narratives could be kept (see a similar approach in ref. [39]).

On the group level, whole-brain analysis (WBA) was performed using the Matlab toolbox SnPM13 (http://warwick.ac.uk/snpm, last checked for updates on Nov. 8, 2019). Such permutation-based, nonparametric method for multiple comparison correction and thresholding is suggested to consistently minimize false positives[105]. Due to the exploratory nature of this study, we used SnPM to avoid false positives that could lead to false interpretation and hypothesis generation. Two group-level models comprising of subject-level contrast images with beta weights of parametric predictors in each GLM (emotional intensity or literariness) averaged over the two story sessions were entered into SnPM for permutation-based one-sample $t$ tests and combined cluster-voxel-extent thresholding (5000 permutations, cluster-forming threshold $p < 0.001$, family-wise error correction $p < 0.05$ on the cluster level).

*Region of interest analysis.* In the current study, we performed an ROI analysis to look at covariation of percent signal change in certain brain areas and behavioural outcomes and individual differences to look at brain−behaviour correlations. Since we had no hypotheses regarding behavioural measures of aesthetic experiences and functional specialization of certain brain areas or networks in aesthetic experiences at this time scale, we performed this analysis at the ROI level. Post-hoc ROIs were based on approximation of the regions found to be significant in the WBA analysis to test relationships between BOLD responses and individual differences in aesthetic and experiential measures for each story and individual differences in reading behaviour and other measures. Based on the results of the WBA (see Fig. 2), we selected ROIs in the left angular gyrus, left supramarginal gyrus, left precuneus, and bilateral middle temporal gyrus for the literariness analysis (see Fig. 5a), and the frontal control network and the frontoparietal attention network for the emotional intensity analysis (see Fig. 5b).

We used anatomical and functional terms to generate meta-analysis maps using Neurosynth's "association tests"[106] (see Table 4 for details) that approximated the regions we found to be sensitive to literariness and emotional intensity in the WBA. The association test reveals brain regions that are *more consistently* activated in studies with the target term than in studies without (see https://neurosynth.org/faq/). All meta-analysis maps were corrected for multiple comparisons at false discovery rate (FDR) $q < 0.01$. We then thresholded the maps using $z$-scores to isolate one reasonably sized cluster around each anatomical region. A custom Matlab script was used to obtain the centre-of-mass coordinates in MNI space of each masked cluster. A spherical ROI of 8 mm radius was then drawn around each centre-of-mass coordinate using Matlab toolbox MarsBaR[107]. For large functional networks (labelled as frontal control network and frontoparietal attention network), we directly used the FDR-corrected meta-analysis maps with an arbitrary cluster-extent threshold of ten voxels as the ROI masks.

Individual participant's percent signal changes (PSC) per story were extracted from ROIs by four GLMs on the single-subject level with the two predictors of interest (emotional intensity and literariness) and two stories (DH and DM). We built these GLMs with an event-related design without the parametric modulation by binarizing the parametric predictors (see Fig. 1c). An event was defined to have the onset and duration of a semantic event which has either an emotional intensity rating strictly over 4 (on a scale of 1–7, with 7 being the most emotional) or a literariness rating strictly over 1.5 (on a scale of 1–2, with 2 being the most literary). All other factors were kept the same as in the WBA. The PSC values per ROI were then correlated with story-specific measures and individual differences measures (see Table 2) using Spearman's rank correlation coefficient $\rho$.

**Reporting summary**. Further information on research design is available in the Nature Research Reporting Summary linked to this article.

## Data availability
The datasets generated and/or analysed during the current study are available on Open Science Framework repository, https://osf.io/w2uad/.

## Code availability
The custom code we used is available on Open Science Framework repository, https://osf.io/w2uad/.

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

## Acknowledgements

This project was supported by the Haverford College KINSC Summer Scholar funding awarded to Y.W., the Nederlandse Wetenschappelijk Onderzoek (NWO-Vidi-276-89-007) awarded to R.W., and the Dolores Smith Innovation fund awarded to Penn Center for Neuroaesthetics. We want to thank three anonymous reviewers for their helpful input on a previous draft.

## Author contributions

F.H. and Y.W. contributed equally to this work. The study concept and experimental design were conceived and developed by F.H., Y.W., and A.C. Data collection and processing were performed by Y.W., M.M., and R.W. Data analysis and visualizations were done by Y.W. and F.H. F.H. and Y.W. drafted the initial manuscript. R.W., M.M., and A.C. contributed to the writing of the manuscript and provided critical feedback. All authors critically contributed to the final version.

## Competing interests

The authors declare no competing interests.
