## [Transparent Peer Review File · Communications Biology]

Reviewers' comments:

Reviewer #1 (Remarks to the Author):

I read with much interest this work, which reports secondary analyses of previously collected neuroimaging data using fresh ratings of literariness and emotional intensity. In the original study, participants undergoing fMRI scanning were asked to read 2 narratives for their own enjoyment. In the current work, the authors collected binary ratings of literariness (stylistically remarkable/well-written vs. normal) and 7-point Likert ratings of emotional intensity (from completely unaffected to felt intense emotion) from two independent groups of participants. The aim of this work was to investigate whether aesthetic appreciation in literary reading is supported by distinct neural system depending on foregrounding vs. backgrounding effects. The authors find support for this distinction: while literariness seems to engage the semantic neural network, felt emotional intensity disengaged the control and attention networks, as if participants could let go of executive control and attentional processes when emotionally engaged by the narrative.

The work is very novel and compelling. It suffers from limitations such as the fact that the fresh ratings are not from the same participant sample that underwent scanning, but it is a very good initial step into investigating how foregrounding and backgrounding may or may not be distinctly represented at the neural level.

I have some suggestions on how to improve the current manuscript. I found it difficult to tease apart what was done in the current study and what was done in the previous study (Hartung et al., 2017), so the method section could perhaps be modified to make these distinction crystal clear. I also have some other suggestions with regards to the presentation of the results and the discussion.

Long abstract:

"We collected word by word ratings..... to predict blood pxygen-level-dependent signal changes in a THIRD SAMPLE OF 52 participants who..."

Emotional intensity

At the end of the introduction, you have no specific predictions for emotional intensity during story engagement. You also repeat this in the discussion, also stating that emotional reactions can either be due to a "physiological emotional state" (more automatic) or an "appraisal" process, which would involve higher-order cortices. You state that "this distinction may explain the absence of subcortical activations typically associated with emotion". Crucially, you find no structures of the emotion neural network activated in response to emotional intensity, neither subcortical nor cortical.

First, you could have predictions for activations in response to emotional intensity irrespective of this distinction. You could clearly state that you don't know whether a more physiological or a more emotion-appraisal response would be elicited, but you can predict which structures would be activated in response to one or the other. Second, the lack of any activations in the emotion neural network in response to emotional intensity needs to be clearly acknowledged and discussed.

If a narrative elicits more appraisal-like processes as an emotional response, I would expect to find activation of the orbito-frontal cortex, but there is nothing there. What's going on? Perhaps participants react spontaneously and automatically to emotional intensity by activating the amygdala, but higher order processes involved in making sense of the narrative perhaps inhibit its activation? There used to be work from Wager and/or Phan that mentioned this. We know that amygdala reacts to evolutionary relevant stimuli when little to no cognitive demand is imposed by the task, while ACC or OFC react to emotion when more cognitively demanding tasks are required. We also no that emotional arousal ratings tend to be less stable across raters compared to emotional valence, os perhaps ratings from an independent group did not mimic emotional responses in a different group reliably enough to find any emotion network active. There is plenty or neuroimaging work out there

showing amygdala and other emotion-areas active in response to natural reading of stories, so the lack of such findings in your study is at odds with previous literature. We also know that arousal and valence have been distinguished at the neural level (Lewis et al 2007 and subsequent work but also work with odours and taste): while valence recruits OFC and ACC, arousal recruits amygdala and insula, so once again there is scope for an inhibition of such responses due to other higher order processes going on. Any of these above are just ideas/suggestions that you are free to consider or reject depending on your own evaluation, but certainly more work is needed in terms of acknowledging and interpreting the lack of emotion-related activations.

Materials & Method

Could you clearly state at the beginning of each of the sub-section whether you are referring to work published in Hartung et al 2017 or to the current work?

Perhaps you could have a main header, under Materials and Method, for all what happened in the previous study and then another main header for all what happened in the current one.

Stories -> "IN A /OUR PREVIOUS STUDY, two Dutch literary short stories..."

Ratings of the stories -> Present study or past study??

Experiential aspects of reading -> did all of this happen in the previous study then?

Individual differences -> previous study?

Reading habits -> current study?

Your Table 1 is very clear. However, in the text the reader gets confused about what was done when and what from the previous and current study has been used in the present work.

fMRI, end of last paragraph: what is "pinkie"? One of the fingers?

Data analysis, behavioural rating survey

There was a low positive correlation between literariness and emotional intensity ($r = .16$). How did you get to the 2.5% of shared variance?

fMRI analyses: could you explain what the pass filter is and why you use it/what it does to the data?

How come you used SnPM?

Figures 4 and 5: could you associate the colours with the regions you mention in the caption? E.g.: left superior parietal lobule (ISLP; in yellow); superior temporal gyrus (in blue), etc.? Very same for Figure 5: is FPAN in blue and FCN in yellow/green?

"We used anatomical and functional terms to generate meta-analysis maps" I found this statement puzzling given my understanding of what a meta-analysis is... you are not analysing data from many different studies. Could you perhaps elaborate a bit more on this?

Beginning of discussion:

Given what participants did in the scanner (reading for pleasure) and what raters did (explicitly evaluation literariness and felt emotional intensity), you'd need to be a bit more careful with your wording: Did you perhaps explore "implicating processing of literary aspects of language and implicit emotional reactions to literary language"? Because you have explicit ratings and hope to find related activity in the brain because, hopefully, the participants undergoing scanning were implicitly reacting to striking phrases and emotive sentences. I find your wording "the neural correlates of engaging with

literary language and emotional intensity when reading narratives" a bit too vague. Also one need to be clear that what was measured was felt emotional intensity, and not the emotional intensity of the portion of text itself, i.e., it's not a rating of how intense that word is but how intensely one felt in response to that word.

Reviewer #2 (Remarks to the Author):

In this paper, the authors report results from parametric whole-brain and ROI analyses of fMRI data collected in a previously reported study. In this study, subjects were scanned while listening to stories, and separately completed several self-report measures of their experience of the narrative and their own dispositions and reading habits. The stories were also rated on emotional intensity and literariness by independent raters.

I found the overall topic very interesting and think that the specific angle of literary vs. emotional processing of narratives is novel and agree with the authors that their findings can lay the foundation for future research. I also want to applaud the authors for their very transparent data sharing and reporting practices.

I had some issues with the framing of the paper and interpretation of the neuroimaging results, and think that the paper would benefit from more engagement with the affective neuroscience literature than it does currently. In general I find myself not very convinced that these results really speak to different brain systems for literary vs. emotional aesthetic engagement with texts per se. For example, one of the conclusions laid out in the introduction is that "aesthetic experiences cannot be reduced to a common neural basis", which I found stated rather strongly - these results offer an early suggestion that literary vs. emotional engagement in literary narratives recruit different brain systems, not really that "aesthetic experiences cannot be reduced to a common neural basis". In general I found the paper very interesting without the need to frame things in terms of aesthetic emotion, which the authors acknowledge is a vague/contentious/poorly operationalized concept.

And it's not clear that aesthetic engagement is the best way to think about this since the regressors model aspects of the text itself, not how the readers are engaged with it - as the authors state in the abstract, "Our results confirm a neural dissociation in processing the literary form and the emotional content of stories", which I agree with and think the results speak to strongly, less so for the subjective aesthetic emotional experience. The behavioral correlations were all fascinating and compelling as well, but there is a substantially more well-developed literature on the neuroscience of emotion to engage with that could help elucidate these findings.

I thought that foregrounding was not an especially useful construct here; on page 6 the authors seem to be setting up a distinction wherein foregrounded vs. backgrounded maps cleanly onto literary vs. emotional aesthetic engagement but I found this a bit muddled and not well-operationalized enough of a construct to help with the interpretation of neuroimaging findings (and some of the interpretations of the individual behavioral correlations overly speculative as a result).

Emotional intensity doesn't seem to correlate positively with anything in particular, but has the strong negative association with activation in the attention network. Parameter estimates for emotional intensity might be noisy - this is seems like a possibility looking at the maps shown in supplementary materials (Supplementary Figure 2). One possibility would be to adjust for more stimulus features in the parametric models. On page 29 the authors state that "low frequency and literariness are indistinguishable in our design since foregrounded language, by definition, occurs less frequently", but this is an empirical question - based on the stimulus samples the authors provide, I'm not confident that the words used in the high literariness passages are necessarily less frequent in the objective sense. Normed frequency ratings are easy to obtain in English, although I'm not sure about Dutch. But

it seems like controlling for some lower-level stimulus features is essential here, honestly, given the high-level interpretations. Length of words in syllables are also commonly adjusted for in fMRI studies involving reading and that could be useful here. There are also databases of normed valence ratings that could help - e.g. affective normed for English words (ANEW) although again I'm not sure about Dutch. Given the high-level interpretations of the findings, these sorts of low-level differences should ideally be accounted for.

Also, the highly rated segments for literariness seem to skew toward a rather negative valence - could just be the examples, but points again to the need to perform more measurement and comparison on the narrative stimuli themselves.

Reviewer #3 (Remarks to the Author):

Review for Communications Biology

The current fMRI study explored the neural correlates of literariness and emotional intensity when listening to narratives. The results found diverse brain activations that are responsible for the literary and emotion intensity. Literary was found to be related to left angular gyrus, left supramarginal gyrus, and precuneus that are associated with semantic processing. By contrast, emotion intensity was related to a bilateral frontoparietal network that is often associated with controlled attention.

The authors provided interesting data to reveal the neural underpinnings of aesthetic processing in literature. However, several concerns should be addressed.

1. The article title was confusing. Out of ordinary? In my view, the literary and emotion intensity were ordinary and nothing is out of ordinary. Please delete. The rest of the title was enough clear, such as, Neural correlates of literary form and emotional content in aesthetic engagement with literature: A fMRI study.

2. The measurement for each scale should provide reliability and validity.

3. The authors need to clarify the 27 participants were enough to rate the related scales. In addition, please proffer emotion granularity of the participants (see related work of Barrett). People are different in emotion granularity, and it is possible that some of your participants were not good at emotion understanding and had poor emotion categories.

4. For tables and figures, when abbreviations were used, if possible, please provide the full name at the caption.

5. The authors were asked to use Bonferroni correction rather than Sidak, since the comparisons were less than 5.

6. The authors said that aesthetic emotions are in debate and the concept is elusive. However, more details should be provided to elucidate this claim.

7. When discuss the frontal parietal areas that were related to emotion and conflict, the authors were asked to review relevant studies that explored this area with emotion words (Kanske, 2012; Kanske & Kotz, 2010, 2011; Wu & Zhang, 2019; Zhang, Teo, & Wu, 2019; Zhang, Wu, Yuan, & Meng, 2019). In fact, emotion has been found to reduce conflict due to its salience and superiority in attention capture. These findings might be helpful to understand the current results.

8. Recent emotion word processing review has been updated since then (Citron, 2012). Please consider add more recent ones (Duñabeitia & García-Palacios, 2019; Hinojosa, Moreno, & Ferré, 2019;

Wu & Zhang, 2020).

9. Why participants were asked to listen to the stories? The authors should provide reasons for this design. Usually, visual reading is more common.

10. When discuss about word frequency effect, the authors are suggested to consult the work from Bysbeart (Brysbaert, Mandera, & Keuleers, 2018). The current version is weak and not informative.

11. The authors should revise parts that are difficult to follow. For instance, "This correlation suggests that immersion in a narrative disengages controlled attention. ", I am confused with this sentence.

Brysbaert, M., Mandera, P., & Keuleers, E. (2018). The word frequency effect in word processing: A review update. *Current directions in psychological science*, 27. doi:10.1177/0963721417727521

Citron, F. M. (2012). Neural correlates of written emotion word processing: a review of recent electrophysiological and hemodynamic neuroimaging studies. *Brain and language*, 122(3), 211-226.

Duñabeitia, J. A., & García-Palacios, A. (2019). The transdisciplinary nature of affective neurolinguistics: a commentary on Hinojosa, Moreno and Ferré (2019). *Language, Cognition and Neuroscience*, 1-3.

Hinojosa, J., Moreno, E., & Ferré, P. (2019). Affective neurolinguistics: towards a framework for reconciling language and emotion. *Language, Cognition and Neuroscience*, 1-27.

Kanske, P. (2012). On the influence of emotion on conflict processing. *Frontiers in Integrative Neuroscience*, 6.

Kanske, P., & Kotz, S. A. (2010). Modulation of early conflict processing: N200 responses to emotional words in a flanker task. *Neuropsychologia*, 48(12), 3661-3664.

Kanske, P., & Kotz, S. A. (2011). Conflict processing is modulated by positive emotion: ERP data from a flanker task. *Behavioural brain research*, 219(2), 382-386.

Wu, C., & Zhang, J. (2019). Conflict Processing is Modulated by Positive Emotion Word Type in Second Language: An ERP Study. *Journal of psycholinguistic research*, 1-14.

Wu, C., & Zhang, J. (2020). Emotion word type should be incorporated in affective neurolinguistics: A commentary on Hinojosa, Moreno and Ferré (2019). *Language, Cognition and Neuroscience*, 35(7), 840-843.

Zhang, J., Teo, T., & Wu, C. (2019). Emotion words modulate early conflict processing in a flanker task: differentiating emotion-label words and emotion-laden words in second language. *Language and Speech*, 62(4), 641-651.

Zhang, J., Wu, C., Yuan, Z., & Meng, Y. (2019). Differentiating emotion-label words and emotion-laden words in emotion conflict: an ERP study. *Experimental Brain Research*, 1-8.

Reviewers' comments:

Reviewer #1 (Remarks to the Author):

I read with much interest this work, which reports secondary analyses of previously collected neuroimaging data using fresh ratings of literariness and emotional intensity. In the original study, participants undergoing fMRI scanning were asked to read 2 narratives for their own enjoyment. In the current work, the authors collected binary ratings of literariness (stylistically remarkable/well-written vs. normal) and 7-point Likert ratings of emotional intensity (from completely unaffected to felt intense emotion) from two independent groups of participants. The aim of this work was to investigate whether aesthetic appreciation in literary reading is supported by distinct neural system depending on foregrounding vs. backgrounding effects. The authors find support for this distinction: while literariness seems to engage the semantic neural network, felt emotional intensity disengaged the control and attention networks, as if participants could let go of executive control and attentional processes when emotionally engaged by the narrative.

The work is very novel and compelling. It suffers from limitations such as the fact that the fresh ratings are not from the same participant sample that underwent scanning, but it is a very good initial step into investigating how foregrounding and backgrounding may or may not be distinctly represented at the neural level.

I have some suggestions on how to improve the current manuscript. I found it difficult to tease apart what was done in the current study and what was done in the previous study (Hartung et al., 2017), so the method section could perhaps be modified to make these distinction crystal clear. I also have some other suggestions with regards to the presentation of the results and the discussion.

Long abstract:

“We collected word by word ratings..... to predict blood oxygen-level-dependent signal changes in a THIRD SAMPLE OF 52 participants who...”

***Response:** We added the suggested phrase. We also revised the Method section for more clarity throughout (see also answer below).*

Emotional intensity

At the end of the introduction, you have no specific predictions for emotional intensity during story engagement. You also repeat this in the discussion, also stating that emotional reactions can either be due to a “physiological emotional state” (more automatic) or an “appraisal” process, which would involve higher-order cortices. You state that “this distinction may explain the absence of subcortical activations typically associated with emotion”. Crucially, you find no structures of the emotion neural network activated in response to emotional intensity, neither subcortical nor cortical.

First, you could have predictions for activations in response to emotional intensity irrespective of this distinction. You could clearly state that you don't know whether a more physiological or a more emotion-appraisal response would be elicited, but you can predict which structures would be activated in response to one or the other. Second, the lack of any activations in the emotion neural network in response to emotional intensity needs to be clearly acknowledged and discussed. If a narrative elicits more appraisal-like processes as an emotional response, I would expect to find activation of the orbito-frontal cortex, but there is nothing there. What's going on? Perhaps participants react spontaneously and automatically to emotional intensity by activating the amygdala, but higher order processes involved in making sense of the narrative perhaps inhibit its activation? There used to be work from Wager and/or Phan that mentioned this. We know that amygdala reacts to evolutionary relevant stimuli when little to no cognitive demand is imposed by the task, while ACC or OFC react to emotion when more cognitively demanding tasks are required. We also note that emotional arousal ratings tend to be less stable across raters compared to emotional valence, or perhaps ratings from an independent group did not mimic emotional responses in a different group reliably enough to find any emotion network active. There is plenty of neuroimaging work out there showing amygdala and other emotion-areas active in response to natural reading of stories, so the lack of such findings in your study is at odds with previous literature. We also know that arousal and valence have been distinguished at the neural level (Lewis et al 2007 and subsequent work but also work with odours and taste): while valence recruits OFC and ACC, arousal recruits amygdala and insula, so once again there is scope for an inhibition of such responses due to other higher order processes going on. Any of these above are just ideas/suggestions that you are free to consider or reject depending on your own evaluation, but certainly more work is needed in terms of acknowledging and interpreting the lack of emotion-related activations.

***Response:** We agree that we can elaborate on possible hypotheses for this predictor without taking a specific position. We now added such hypotheses based on previous work on narrative suspense. We further added two paragraphs to the discussion addressing the lack of positive activations and speculate why this absence of finding might be the case. Your comment inspired us to think more about the meaning of this aspect of our study, also in response to the studies that previously found effects in primary emotion areas. After looking into these studies, most studies used popular or enculturated narratives that might elicit different emotional responses than our unfamiliar rather high-brow literary stories. We added a point to the discussion regarding this important distinction and we hope you think this point is as relevant as we do.*

Materials & Method

Could you clearly state at the beginning of each of the sub-section whether you are referring to work published in Hartung et al 2017 or to the current work?

Perhaps you could have a main header, under Materials and Method, for all what happened in the previous study and then another main header for all what happened in the current one.

Stories -> "IN A /OUR PREVIOUS STUDY, two Dutch literary short stories..."

Ratings of the stories -> Present study or past study??

Experiential aspects of reading -> did all of this happen in the previous study then?

Individual differences -> previous study?

Reading habits -> current study?

Your Table 1 is very clear. However, in the text the reader gets confused about what was done when and what from the previous and current study has been used in the present work.

Response: *We have added highlighted pointers in the first line of each section in the Methods to indicate whether the data collection and/or analysis was done in Hartung et al. 2017 or the present study. We also added clarification in potentially ambiguous cases.*

fMRI, end of last paragraph: what is “pinkie”? One of the fingers?

Response: *It should be ‘little finger’, instead of pinkie which we corrected.*

Data analysis, behavioural rating survey

There was a low positive correlation between literariness and emotional intensity ($r = .16$). How did you get to the 2.5% of shared variance?

Response: *The coefficient of determination (R^2) was calculated by taking the square of Pearson’s r (0.157). This R^2 indicates the amount of variance in emotional intensity rating (dependent variable) that can be accounted for by the literariness rating (arbitrarily chosen as the independent variable in our code), and vice versa (i.e. emotional intensity rating as independent variable and literariness rating as dependent variable).*

While the positive correlations are low, we do not think they are spurious. We informally ran simulations correlating completely random mean literariness ratings (drawing from uniform distribution between 1(normal) and 2 (stylistically remarkable)) with emotional intensity.

Correlations were not close to the correlation we obtained in the study. It could mean that readers when emotionally aroused are more sensitive to literariness or writers are inclined to sometimes use literary devices at moments of emotional arousal. Having said that, given the low correlations, we are reluctant to insert these speculations in the paper.

fMRI analyses: could you explain what the pass filter is and why you use it/what it does to the data?

Response: *A high-pass filter is generally used to process BOLD signals to decrease the intrinsic low frequency “drifts,” or noises, in fMRI and thus to improve the signal-to-noise ratio. By setting a cutoff period (usually 128s, the default in SPM package), signals with a period shorter than 128s are preserved but signals with a period longer than 128s are attenuated. In our study, the most prominent fluctuation in emotional intensity has a period longer than 128s, which would be filtered out with the default cutoff period. We conducted a fast Fourier transform to analyze the power*

spectrum of our predictors (averaged emotional intensity ratings over time for each story) to determine the exact period of this prominent fluctuation. We then adjusted the highpass filter size to 256s which includes the stronger signal relevant for emotion changes, but also filters out as much lower frequency noises as possible.

Friston, K. J. (Ed.). (2007). *Statistical parametric mapping: The analysis of functional brain images (1st ed)*. Chapter 14, pg 183-184. Elsevier/Academic Press.

How come you used SnPM?

Response: We decided to use SnPM because non-parametric testing tends to be more robust against type 1 error, especially if variables are correlated or if there is potential noise in the data. Since this study is exploratory in nature we wanted to avoid having false positives that might lead to false interpretations and hypothesis generation. So instead, we report cluster thresholding with a more conservative threshold and add a more lenient thresholding analysis in the supplementary materials (see S2).

While the ideal method for multiple comparison correction and thresholding in fMRI clusters continues to be debated (e.g., Kessler et al., 2017; Slotnick 2017; Vandekar et al., 2019; Ostwald et al., 2019), the seminal paper that tested the false positive rate using different methods and parameters (Eklund et al., 2016) suggested that permutation-based, nonparametric methods on the group level performed consistently well. We therefore used SnPM, which is an extension package to SPM that enables permutation-based cluster-extent thresholding on the group level (Nichols & Holmes, 2002; Winkler et al., 2014).

Kessler, D., Angstadt, M., & Sripada, C. S. (2017). Reevaluating “cluster failure” in fMRI using nonparametric control of the false discovery rate. *Proceedings of the National Academy of Sciences*, 114(17), E3372–E3373. <https://doi.org/10.1073/pnas.1614502114>

Slotnick, S. D. (2017). Cluster success: FMRI inferences for spatial extent have acceptable false-positive rates. *Cognitive Neuroscience*, 8(3), 150–155. <https://doi.org/10.1080/17588928.2017.1319350>

Vandekar, S. N., Satterthwaite, T. D., Xia, C. H., Adebimpe, A., Ruparel, K., Gur, R. C., Gur, R. E., & Shinohara, R. T. (2019). Robust spatial extent inference with a semiparametric bootstrap joint inference procedure. *Biometrics*, 75(4), 1145–1155. <https://doi.org/10.1111/biom.13114>

Ostwald, D., Schneider, S., Bruckner, R., & Horvath, L. (2019). Power, positive predictive value, and sample size calculations for random field theory-based fMRI inference. *BioRxiv*, 613331. <https://doi.org/10.1101/613331>

Eklund, A., Nichols, T. E., & Knutsson, H. (2016). Cluster failure: Why fMRI inferences for spatial extent have inflated false-positive rates. *Proceedings of the National Academy of Sciences*, 113(28), 7900–7905. <https://doi.org/10.1073/pnas.1602413113>

Nichols, T. E., & Holmes, A. P. (2002). Nonparametric permutation tests for functional neuroimaging: A primer with examples. *Human Brain Mapping*, 15(1), 1–25. <https://doi.org/10.1002/hbm.1058>

Winkler, A. M., Ridgway, G. R., Webster, M. A., Smith, S. M., & Nichols, T. E. (2014). Permutation inference for the general linear model. *NeuroImage*, 92, 381–397. <https://doi.org/10.1016/j.neuroimage.2014.01.060>

Figures 4 and 5: could you associate the colours with the regions you mention in the caption? E.g.: left superior parietal lobule (ISLP; in yellow); superior temporal gyrus (in blue), etc.? Very same for Figure 5: is FPAN in blue and FCN in yellow/green?

Response: Done.

“We used anatomical and functional terms to generate meta-analysis maps” I found this statement puzzling given my understanding of what a meta-analysis is... you are not analysing data from many different studies. Could you perhaps elaborate a bit more on this?

Response: *Neurosynth is a database of fMRI studies with an automated tool that can search studies based on terms or activation coordinates, and generate metaanalysis images by pooling together relevant studies (Yarkoni et al., 2011). We searched for terms such as “angular gyrus” and “control” and generated activation maps based on the “association test” function. The association test reveals brain regions that are more consistently activated in studies with the target term than in studies without (see <https://neurosynth.org/faq/>). By using these metaanalysis maps that approximated the regions we found in WBA (instead of directly using the WBA regions), we tried to avoid “double dipping” (circular analysis) that would distort results (see Kriegeskorte et al., 2009). Because of the limitation of our existing data, better methods such as split-sample cross validation is not feasible (the split-half sample size would be too small), but we consider our brain-behavior ROI correlations valuable first steps in building hypotheses for subsequent, hypothesis-testing studies.*

*Kriegeskorte, N., Simmons, W. K., Bellgowan, P. S., & Baker, C. I. (2009). Circular analysis in systems neuroscience – the dangers of double dipping. *Nature Neuroscience*, 12(5), 535–540.*

<https://doi.org/10.1038/nn.2303>

*Yarkoni, T., Poldrack, R. A., Nichols, T. E., Van Essen, D. C., & Wager, T. D. (2011). Large-scale automated synthesis of human functional neuroimaging data. *Nature Methods*, 8(8), 665–670.*

<https://doi.org/10.1038/nmeth.1635>

Beginning of discussion:

Given what participants did in the scanner (reading for pleasure) and what raters did (explicitly evaluation literariness and felt emotional intensity), you’d need to be a bit more careful with your wording: Did you perhaps explore “implicating processing of literary aspects of language and implicit emotional reactions to literary language”? Because you have explicit ratings and hope to find related activity in the brain because, hopefully, the participants undergoing scanning were implicitly reacting to striking phrases and emotive sentences. I find your wording “the neural correlates of engaging with literary language and emotional intensity when reading narratives” a bit too vague. Also one need to be clear that what was measured was felt emotional intensity, and not the emotional

intensity of the portion of text itself, i.e., it's not a rating of how intense that word is but how intensely one felt in response to that word.

***Response:** This is an important point, which we now account for in edits at the beginning of our discussion. Thank you for pointing this issue out!*

Reviewer #2 (Remarks to the Author):

In this paper, the authors report results from parametric whole-brain and ROI analyses of fMRI data collected in a previously reported study. In this study, subjects were scanned while listening to stories, and separately completed several self-report measures of their experience of the narrative and their own dispositions and reading habits. The stories were also rated on emotional intensity and literariness by independent raters.

I found the overall topic very interesting and think that the specific angle of literary vs. emotional processing of narratives is novel and agree with the authors that their findings can lay the foundation for future research. I also want to applaud the authors for their very transparent data sharing and reporting practices.

I had some issues with the framing of the paper and interpretation of the neuroimaging results, and think that the paper would benefit from more engagement with the affective neuroscience literature than it does currently. In general I find myself not very convinced that these results really speak to different brain systems for literary vs. emotional aesthetic engagement with texts per se. For example, one of the conclusions laid out in the introduction is that “aesthetic experiences cannot be reduced to a common neural basis”, which I found stated rather strongly - these results offer an early suggestion that literary vs. emotional engagement in literary narratives recruit different brain systems, not really that "aesthetic experiences cannot be reduced to a common neural basis". In general I found the paper very interesting without the need to frame things in terms of aesthetic emotion, which the authors acknowledge is a vague/contentious/poorly operationalized concept.

***Response:** We substantially added to our theoretical clarification and added some in depth review of related studies on affective responses to language. We also toned down our conclusions as we agree with you that these comments were too strongly worded.*

And it's not clear that aesthetic engagement is the best way to think about this since the regressors model aspects of the text itself, not how the readers are engaged with it - as the authors state in the abstract, "Our results confirm a neural dissociation in processing the literary form and the emotional content of stories", which I agree with and think the results speak to strongly, less so for the subjective aesthetic emotional experience. The behavioral correlations were all fascinating and

compelling as well, but there is a substantially more well-developed literature on the neuroscience of emotion to engage with that could help elucidate these findings.

Response: *We added substantially to the theoretical framing of the study and elaborate on the concepts of aesthetic emotions, foregrounding, emotional intensity as measured here, as well as its relation to emotion and affective neurolinguistics. We also added several studies addressing suspense and intensity, as well as arousal when people are engaged with stories.*

I thought that foregrounding was not an especially useful construct here; on page 6 the authors seem to be setting up a distinction wherein foregrounded vs. backgrounded maps cleanly onto literary vs. emotional aesthetic engagement but I found this a bit muddled and not well-operationalized enough of a construct to help with the interpretation of neuroimaging findings (and some of the interpretations of the individual behavioral correlations overly speculative as a result).

Emotional intensity doesn't seem to correlate positively with anything in particular, but has the strong negative association with activation in the attention network. Parameter estimates for emotional intensity might be noisy - this seems like a possibility looking at the maps shown in supplementary materials (Supplementary Figure 2). One possibility would be to adjust for more stimulus features in the parametric models. On page 29 the authors state that “low frequency and literariness are indistinguishable in our design since foregrounded language, by definition, occurs less frequently”, but this is an empirical question - based on the stimulus samples the authors provide, I’m not confident that the words used in the high literariness passages are necessarily less frequent in the objective sense. Normed frequency ratings are easy to obtain in English, although I’m not sure about Dutch. But it seems like controlling for some lower-level stimulus features is essential here, honestly, given the high-level interpretations. Length of words in syllables are also commonly adjusted for in fMRI studies involving reading and that could be useful here. There are also databases of normed valence ratings that could help - e.g. affective normed for English words (ANEW) although again I’m not sure about Dutch. Given the high-level interpretations of the findings, these sorts of low-level differences should ideally be accounted for.

Response: *We revised the conceptual framing substantially. Lexical frequency is orthogonal to foregrounding. Foregrounding is defined as probability within a given context. With respect to foregrounding in a literary story, the story itself (or potentially related texts, e.g. fan fiction or stories by the same author or within the same universe) and the reading situation is the only context. In fact, it is often the case that particularly frequent lexical items are foregrounded in a story. A common example of this is when colloquial language in literary writing is used to contrast a formal narrative style. Foregrounding is also not restricted to any linguistic level or units (e.g. lexical items/words or sentences); rather it is a subjective experience of salience of the literary form in parsing intentionally constructed language/narratives. This is not to say that it is not an interesting research question in itself to test in how far “context-free” word ratings relate to perceived foregroundedness in literary texts, but this is a new research question that is beyond the scope of this paper.*

Since there are no objective measures to test for such text sequences, we decided to assess this with subjective ratings of a group of naive participants and then test for the reliability of their subjective ratings. As you can see from the intraclass correlation coefficient the cross-subject reliability for foregrounding is quite high indicating that the overlap of text sequences perceived as foregrounded is substantial and that this rating is a robust measure of experienced salience of the literary form. Similarly, emotion ratings for words cannot reliably capture emotional content in narratives. We added a theoretical model that explains and elaborates this point based on examples. There is some noise in this regressor as the intensity ratings are not from the same subjects as the neural correlations. We agree that confirmatory experiments are needed to take individual variation into account. However, our findings actually map well on theoretical models of suspense and narrative engagement and relate to similar findings from visual narratives and non-literary narratives. We added these studies to the discussion.

We realise that not having made this conceptual distinction clearer is a shortcoming of our manuscript. To address this shortcoming, we added an in depth explanation in the introduction when introducing the concepts of foregrounding and intensity and added more background literature on related concepts.

Also, the highly rated segments for literariness seem to skew toward a rather negative valence - could just be the examples, but points again to the need to perform more measurement and comparison on the narrative stimuli themselves.

Response: *This observation is correct and in fact well represents the content of the narratives. Both narratives almost exclusively contain negatively valenced segments as both stories are placed in gloomy and mysterious settings and revolve around a threatening situation for their protagonist. Both stories are literary fiction with several high-brow elements. De Muur is about an elderly woman who suffers from paranoia and hallucinations. The story is told from her perspective how she tries to survive a night with the man next door trying to attack her. De Mexican dog is placed in the 1930s about a (Jewish) child who encounters a situation they cannot control and is forced into a self-constructed headphone mount by a classmate's father that emits a bone chilling noise. This situation is later related to a memory of the child years later first encountering of Hitler's voice on the radio. Because both stories are dominantly negatively valenced, we did not investigate valence as a predictor and instead only used emotional intensity which relates to suspense and arousal.*

Reviewer #3 (Remarks to the Author):

Review for Communications Biology

The current fMRI study explored the neural correlates of literariness and emotional intensity when listening to narratives. The results found diverse brain activations that are responsible for the literary

and emotion intensity. Literary was found to be related to left angular gyrus, left supramarginal gyrus, and precuneus that are associated with semantic processing. By contrast, emotion intensity was related to a bilateral frontoparietal network that is often associated with controlled attention.

The authors provided interesting data to reveal the neural underpinnings of aesthetic processing in literature. However, several concerns should be addressed.

Response: *First of all, we want to thank you for the thoughtful and constructive comments on our manuscript. We incorporated many of them and improved on the terminology and conceptual clarifications in response to your comments.*

The concepts we were interested in in this study are 1) global modulation of intensity or arousal over time during narrative engagement, which is related to suspense and focus/attention, and 2) foregrounding, which in the context of literary narratives often translates into literariness. Both arousal and foregrounding can be elusive, highly abstract concepts for lay people, so when measuring the appraisals of both arousal and foregrounding from the groups of lay subjects, we used lay translations which captured these concepts and matched the target language Dutch well (“stylistically remarkable/well-written/standing out” for foregrounded, literary parts; a 7-point Likert scale from “completely unaffected” to “felt intense emotion” for levels of arousal, see page 14 in manuscript). It is important to point out that we were only interested in global modulation/fluctuation in appraisals of the degree/levels of foregrounding and arousal over the 2 complete literary narratives, in order to test for associations with certain brain structures. With the current design we have no means to assess individual categories of emotions and/or the subjective valence to the individual, hence this aspect of emotion processing is beyond the scope of this paper. We think that word-wise assessments of such categories as are often used for word and sentence processing measures are not able to reliably reflect these categories on the level of a literary narrative as the context (both of the ‘reading fiction situation’ as well as the closed world of the text itself) can significantly distort these features compared to prototypical word use. Similarly, word frequency and other psycholinguistic variables will not conceptually capture foregrounding. We explain this in detail below under your respective comments.

We realise that it was a shortcoming in our original manuscript that we did not conceptually disentangle these concepts in the introduction and now justify our operationalization of these concepts in more detail. We revised and added at several points clarification and justifications.

1. The article title was confusing. Out of ordinary? In my view, the literary and emotion intensity were ordinary and nothing is out of ordinary. Please delete. The rest of the title was enough clear, such as, Neural correlates of literary form and emotional content in aesthetic engagement with literature: A fMRI study.

Response: *We changed the title for clarity.*

2. The measurement for each scale should provide reliability and validity.

Response:

Done. For word-by-word emotional intensity and literariness ratings - see page 27. For story-specific questionnaires - see page 19

3. The authors need to clarify the 27 participants were enough to rate the related scales. In addition, please proffer emotion granularity of the participants (see related work of Barrett). People are different in emotion granularity, and it is possible that some of your participants were not good at emotion understanding and had poor emotion categories.

Response: *It is indeed an empirical question whether people even agree on emotional intensity and literariness. Since no similar work had been published and since we cannot conduct power calculations for such norming studies, we could not define an appropriate sample size beforehand and instead collected responses from a sample and then tested for interrater reliability (intraclass correlation coefficient) as a proof of concept for the method. When collecting these data, we were aware that this procedure might not work at all. After we collected the rating data and determined internal reliability using intraclass correlation, we were confident to proceed with the fMRI models given the high intraclass correlation which meant that people did in fact agree well above chance both for arousal (emotional intensity) as well as for foregrounding (literariness). We did not measure appraisals of individual emotions, granularity of different emotion categories or types does not apply here. We only measured overall appraisals of modulations in intensity (=arousal) and intraclass correlations show high robustness across subjects meaning that individual differences in granularity of emotion do not affect overall appraisals of intensity.*

4. For tables and figures, when abbreviations were used, if possible, please provide the full name at the caption.

Response: *We changed abbreviations where possible and added a full explanation of each abbreviation where it would be too cumbersome. We also switched out Figure 10, 12, and 13 with new abbreviation free labels.*

5. The authors were asked to use Bonferroni correction rather than Sidak, since the comparisons were less than 5.

Response: *Done.*

6. The authors said that aesthetic emotions are in debate and the concept is elusive. However, more details should be provided to elucidate this claim.

Response: *We elaborate more on the concept of aesthetic emotions now and added several references to relevant opinion and review papers, including:*

Fingerhut, J., & Prinz, J. J. (2020). Aesthetic Emotions Reconsidered. *The Monist*, 103(2), 223–239. <https://doi.org/10.1093/monist/onz037>

Menninghaus, W., Wagner, V., Wassiliwizky, E., Schindler, I., Hanich, J., Jacobsen, T., & Koelsch, S. (2019). What are aesthetic emotions? *Psychological Review*, 126(2), 171–195. <https://doi.org/10.1037/rev0000135>

Wassiliwizky, E. & Menninghaus, W. (2021). Why and How Should Cognitive Science Care about Aesthetics? *Trends in Cognitive Science*, 20:6.

Skov, M., & Nadal, M. (2020). There are no aesthetic emotions: Comment on Menninghaus et al. (2019). *Psychological Review*, 127(4), 640–649. <https://doi.org/10.1037/rev0000187>

7. When discuss the frontal parietal areas that were related to emotion and conflict, the authors were asked to review relevant studies that explored this area with emotion words (Kanske, 2012; Kanske & Kotz, 2010, 2011; Wu & Zhang, 2019; Zhang, Teo, & Wu, 2019; Zhang, Wu, Yuan, & Meng, 2019). In fact, emotion has been found to reduce conflict due to its salience and superiority in attention capture. These findings might be helpful to understand the current results.

Response: *Thank you for the comment and references. We read with interest the referenced affective neurolinguistic studies regarding emotionally valenced words and appreciate this body of evidence that hopefully can lead to a fuller picture of affective language processing from lexical to narrative levels. However, we do not think that conflict as used in the referenced studies (word-level flanker tasks) readily lends itself to our research question and design - there is a significant conceptually unexplored space between resolving conflicting semantic/emotional valence information between word (or even sentence) level processing, and understanding and affectively appreciating conflicts among characters' personalities and desires in a literary story. The content in the stories we used in this experiment does not easily translate into cognitive conflict and we cannot operationalize this post hoc.*

Similarly, word level emotion ratings do not apply to our research goal. Emotions in narratives are complex and cannot be modelled by word level semantic and affective ratings. The Affective Language Comprehension Model (Van Berkum 2018, 2019) addresses this conceptual disjoint in detail. Lexical level emotion ratings of words have little explanatory value for narrative levels. For instance, in one of our stories “The Mexican dog”, the protagonist is repulsed by the fish smell of another character’s hands and terrified by his laughter; the climactic scary situation is built around a pair of headphones. None of the lexical items linked to fish, smell, laughter, or headphones would predict the (negatively valenced) high arousal that is spun around this situation. And because these word level effects do not scale up to embedded, situated language usage, especially in fiction, we did not use word level linguistic measures to model the functional MRI signal.

As a side note, the areas associated in affective processing of word meaning in the referenced papers are substantially more frontal than the deactivations in the largely parietal areas in our study. The only areas in which they overlap are the precuneus, right dorsolateral inferior frontal cortex, and posterior cingulate, which are areas that are not specific to affective processing but also associated

with the default mode network and attention processes. We are not inclined to make the reverse inference to make an association with emotional processing and conflict resolution, especially when the effects go in opposite directions.

In this study we found effects in brain networks linked to attention regulation and possibly inhibition in correlation to the arousal (or emotional intensity) predictor and then generated a reverse inference hypothesis that suspense and intensity during narrative engagement might modulate attentional processes. We wish not to overinterpret this finding in an exploratory study and none of the relevant variables were manipulated or at least controlled for this purpose. The goal was to generate new hypotheses that capture narrative processing better than word and sentence level models and our experiment was not designed to test existing hypotheses.

van Berkum, JJA, (2019). Language Comprehension and Emotion: Where Are the Interfaces, and Who Cares? The Oxford Handbook of Neurolinguistics; Edited by Greig I. de Zubicaray and Niels O. Schiller; DOI: 10.1093/oxfordhb/9780190672027.013.29

8. Recent emotion word processing review has been updated since then (Citron, 2012). Please consider add more recent ones (Duñabeitia & García-Palacios, 2019; Hinojosa, Moreno, & Ferré, 2019; Wu & Zhang, 2020).

Response: *Thank you for pointing us to these more recent reviews and comments. We incorporated some of them in our literature review.*

9. Why participants were asked to listen to the stories? The authors should provide reasons for this design. Usually, visual reading is more common.

Response: *The auditory presentation of the stories in the older study from which we reanalysed the data was chosen for experimental design and fMRI data analysis reasons. With auditory presentation, the onset times and durations for language elements can be held constant across subjects for the entire narrative. Hence, brain activations can easily be related to linguistic input for each subject in a group analysis without having to relate each individual word onset to each participant's scans. Because of this advantage, auditory presentation of longer language segments such as narratives is a common choice in experimental fMRI designs. While it was previously argued that this might be a less natural way to engage with fiction, the recent popularity of audio books and podcasts is a strong counter argument. Visual presentation of narratives is possible in the fMRI scanner (e.g. with tracking eye movements or self-paced reading) but comes with many challenges, such as increased movement of subjects and higher fatigue rates, both factors that affect data quality negatively. Self-paced reading paradigms additionally are highly unnatural. In the current study, we re-used the audio recording to acquire additional ratings in order to keep the new rating studies as close as possible to the original experimental design.*

We added a section explaining this on page 7 under 'Stories'.

10. When discuss about word frequency effect, the authors are suggested to consult the work from Bysbaert (Brysbaert, Mandera, & Keuleers, 2018). The current version is weak and not informative.

Response: *This is a misunderstanding we addressed in the revised draft where we were not clear in the previous version. Lexical frequency is orthogonal to the concepts we are interested in, hence measures of lexical frequency are not useful (see also Van Berkum Affective Language Comprehension Model 2018/2019). The concept of foregrounding is defined as probability within a given context. With respect to foregrounding in a literary story, the story itself (or potentially related texts, e.g. fan fiction or stories by the same author or within the same universe) and the reading situation is the relevant context. In fact, it is often the case that particularly frequent lexical items are foregrounded in a story, e.g. when colloquial language in literary writing is used to contrast the more formal narrative style.*

Foregrounding is also not restricted to any linguistic level or units (e.g. lexical items/words or sentences) rather it is a subjective experience of salience of the literary form in parsing intentionally constructed language/narratives. It occurs at all levels of language and transcends those linguistic categories often within one foregrounded segment. It is hence impossible to classify what type of foregrounding people are sensitive to in a natural text that has not specifically been constructed to disentangle levels of foregrounding, even though people overwhelmingly agree on the cores of foregrounded segments. Since we are not aware of objective measures to test for such text sequences, we decided to assess foregrounding with subjective ratings of a group of naive participants and then test for the reliability of their subjective ratings. As you can see from the intraclass correlation coefficient the cross-subject reliability for foregrounding is high indicating that the overlap of text sequences perceived as foregrounded is quite substantial and that this rating is a robust measure of experienced salience of the literary form.

We realise that this shortcoming of our manuscript was not made clear in the initially submitted draft. To address this shortcoming, we added an in depth explanation with examples in the introduction when introducing the concept of foregrounding on page 4-5.

11. The authors should revise parts that are difficult to follow. For instance, “This correlation suggests that immersion in a narrative disengages controlled attention. “, I am confused with this sentence.

Response: *Edited as suggested.*

Brysbaert, M., Mandera, P., & Keuleers, E. (2018). The word frequency effect in word processing: A review update. *Current directions in psychological science*, 27. doi:10.1177/0963721417727521
Citron, F. M. (2012). Neural correlates of written emotion word processing: a review of recent electrophysiological and hemodynamic neuroimaging studies. *Brain and language*, 122(3), 211-226.
Duñabeitia, J. A., & García-Palacios, A. (2019). The transdisciplinary nature of affective neurolinguistics: a commentary on Hinojosa, Moreno and Ferré (2019). *Language, Cognition and Neuroscience*, 1-3.

- Hinojosa, J., Moreno, E., & Ferré, P. (2019). Affective neurolinguistics: towards a framework for reconciling language and emotion. *Language, Cognition and Neuroscience*, 1-27.
- Kanske, P. (2012). On the influence of emotion on conflict processing. *Frontiers in Integrative Neuroscience*, 6.
- Kanske, P., & Kotz, S. A. (2010). Modulation of early conflict processing: N200 responses to emotional words in a flanker task. *Neuropsychologia*, 48(12), 3661-3664.
- Kanske, P., & Kotz, S. A. (2011). Conflict processing is modulated by positive emotion: ERP data from a flanker task. *Behavioural brain research*, 219(2), 382-386.
- Wu, C., & Zhang, J. (2019). Conflict Processing is Modulated by Positive Emotion Word Type in Second Language: An ERP Study. *Journal of psycholinguistic research*, 1-14.
- Wu, C., & Zhang, J. (2020). Emotion word type should be incorporated in affective neurolinguistics: A commentary on Hinojosa, Moreno and Ferré (2019). *Language, Cognition and Neuroscience*, 35(7), 840-843.
- Zhang, J., Teo, T., & Wu, C. (2019). Emotion words modulate early conflict processing in a flanker task: differentiating emotion-label words and emotion-laden words in second language. *Language and Speech*, 62(4), 641-651.
- Zhang, J., Wu, C., Yuan, Z., & Meng, Y. (2019). Differentiating emotion-label words and emotion-laden words in emotion conflict: an ERP study. *Experimental Brain Research*, 1-8.

REVIEWERS' COMMENTS:

Reviewer #1 (Remarks to the Author):

I am happy with the authors' responses to my questions and with how they addressed or considered some of my comments in the manuscript. However, some of the questions I asked may be relevant to other readers as well and I invite the authors to provide a short rationale for their methodological choices in the manuscript directly, so that the readership can make sense out of them.

Could you explain in the manuscript why you chose a different high-pass filter than the default one?

Could you provide a short and clear rationale for using SnPM?

The part added on Neurosynth with specifications in Table 3 is now clear to me.

Reviewer #2 (Remarks to the Author):

I found that the revised manuscript addressed all my concerns and I find the paper a very worthwhile addition to the literature with its revised conceptual framing, and would recommend it for publication in Communications Biology.

Reviewer #3 (Remarks to the Author):

I don't require any further changes' or something to this effect.

Responses to the remaining Reviewer comments:

Reviewer #1 (Remarks to the Author):

I am happy with the authors' responses to my questions and with how they addressed or considered some of my comments in the manuscript. However, some of the questions I asked may be relevant to other readers as well and I invite the authors to provide a short rationale for their methodological choices in the manuscript directly, so that the readership can make sense out of them.

Could you explain in the manuscript why you chose a different high-pass filter than the default one?

Explanation added for high-pass filter use and how we determined the filter size.

Could you provide a short and clear rationale for using SnPM?

Rationale added for using SnPM.

The part added on Neurosynth with specifications in Table 3 is now clear to me.

Reviewer #2 (Remarks to the Author):

I found that the revised manuscript addressed all my concerns and I find the paper a very worthwhile addition to the literature with its revised conceptual framing, and would recommend it for publication in Communications Biology.

Reviewer #3 (Remarks to the Author):

I don't require any further changes' or something to this effect.